# Enhancing multiphoton upconversion through interfacial energy transfer in multilayered nanoparticles

Bin Zhou[1,4], Bing Tang[1,4], Chuang Zhang[2], Changyun Qin[1], Zhanjun Gu [3], Ying Ma [1✉], Tianyou Zhai [1] & Jiannian Yao[2]

Photon upconversion in lanthanide-doped upconversion nanoparticles offers a wide variety of applications including deep-tissue biophotonics. However, the upconversion luminescence and efficiency, especially involving multiple photons, is still limited by the concentration quenching effect. Here, we demonstrate a multilayered core-shell-shell structure for lanthanide doped $NaYF_4$, where $Er^{3+}$ activators and $Yb^{3+}$ sensitizers are spatially separated, which can enhance the multiphoton emission from $Er^{3+}$ by 100-fold compared with the multiphoton emission from canonical core-shell nanocrystals. This difference is due to the excitation energy transfer at the interface between activator core and sensitizer shell being unexpectedly efficient, as revealed by the structural and temperature dependence of the multiphoton upconversion luminescence. Therefore, the concentration quenching is suppressed via alleviation of cross-relaxation between the activator and the sensitizer, resulting in a high quantum yield of up to 6.34% for this layered structure. These findings will enable versatile design of multiphoton upconverting nanoparticles overcoming the conventional limitation.

[1] State Key Laboratory of Material Processing and Die & Mould Technology, School of Materials Science and Engineering, Huazhong University of Science and Technology, 430074 Wuhan, China. [2] Beijing National Laboratory for Molecular Science, Key Laboratory of Photochemistry, Institute of Chemistry, Chinese Academy of Sciences, 100190 Beijing, China. [3] Key Laboratory for Biomedical Effects of Nanomaterials and Nanosafety, Institute of High Energy Physics and National Center for Nanosciences and Technology, Chinese Academy of Sciences, 100049 Beijing, China. [4] These authors contributed equally: Bin Zhou, Bing Tang ✉email: yingma@hust.edu.cn

The upconversion process involving multistep absorption of two or more low-energy photons to generate a high-energy photon in lanthanide-doped upconversion nanoparticles (UCNPs) enables promising applications in various fields, such as display[1], microlaser[2], solar cell[3], deep-tissue biophotonics[4–6] and super-resolution nanoscopy[7]. Typically, $Yb^{3+}$ ions are doped into these nanoparticles as sensitizer ions with the purpose of transferring energy to activator ions ($Er^{3+}$, $Tm^{3+}$), thereby producing an efficient upconversion[8–10]. However, the brightness and upconverting efficiency of these UCNPs are still limited due to the relatively low doping concentrations of the sensitizer and activator ions. Thus, a direct way to improve their brightness is to increase the concentration of the dopants since the upconversion efficiency is largely dependent on the dopant concentration. Unfortunately, concentration quenching occurs in heavily doped UCNPs because nonradiative energy losses, including energy migration-induced surface quenching and cross-relaxation between neighboring dopant ions, will dominate in this case[11,12].

Recently, substantial efforts have been made to overcome the above obstacles and to enhance the upconversion luminescence (UCL) for lanthanide-doped UCNPs. Inert shell passivation has been demonstrated to be successful in overcoming energy migration-induced surface quenching and enhancing emission intensity in UCNPs[9,12–17]. Energy back-transfer from the activator ions to sensitizer ions can be efficiently blocked in a sandwich structured UCNPs, yielding bright UCL[18]. Wang et al. revealed that a $KYb_2F_7$ host is favorable for multiphoton upconversion of $Er^{3+}$ by minimizing the migration of excitation energy to the defects[10]. High-irradiance excitation is an alternative method that can be used to overcome concentration quenching by enriching the excitation energy[4,19,20]. In the meantime, both brightness enhancement and the promotion of upconverting efficiency has been achieved by construction of dye-UCNP hybrids[21]. Even though these existing approaches are effective for large emission enhancement and for the promotion of upconverting efficiencies, the intrinsic cross-relaxation energy loss in UCNPs has not yet been properly addressed, even though it is a key factor that contributes primarily to concentration quenching when surface quenching is negligible.

Here, we propose that cross-relaxation between dopant ions can be effectively suppressed in UCNPs by combining energy migration and interfacial energy transfer via multilayered structure design, as shown in Fig. 1a. In this structure, the separated location for the $Er^{3+}$ and $Yb^{3+}$ ions in the core and neighboring shell can alleviate $Yb^{3+}$-$Er^{3+}$ cross-relaxation (backward energy transfer) and assure efficient energy transfer at the core-shell interface due to the short $Yb^{3+}$-$Er^{3+}$ distance therein[11,18,22,23]. The inert shell herein can mitigate migration-induced surface quenching and promote energy transfer from $Yb^{3+}$ to $Er^{3+}$ ions. This multilayered structure enables us to overcome the concentration limitation and enrich excitation energy by thickening the sensitizer layer to enhance the UCL, especially at shorter wavelengths. The optimum α-NaYF$_4$:10% Er@NaYbF$_4$ @NaYF$_4$ nanoparticles show an upconversion quantum yield (QY) of 6.34% at an excitation power density of 4.5 W cm$^{-2}$, with a 100-fold enhancement for three-photon upconversion compared with canonical NaY$_{0.78}$F$_4$:Yb$_{0.2}$Er$_{0.02}$@NaYF$_4$ core-shell nanocrystals.

## Results

**Synthesis and brighter UCL of the trilayered UCNPs.** We synthesized both the canonical α-NaYF$_4$:Yb,Er@NaYF$_4$ (C$_Y$:20% Yb, 2% Er@S$_Y$) core-shell (Fig. 1b and Supplementary Fig. 1) and α-NaYF$_4$:Er@NaYbF$_4$@NaYF$_4$ (C$_Y$:Er@S$_{Yb}$@S$_Y$) core-shell-shell nanoparticles by an epitaxial growth method (Fig. 1a and Supplementary Fig. 2). The crystal phase and high quality of these

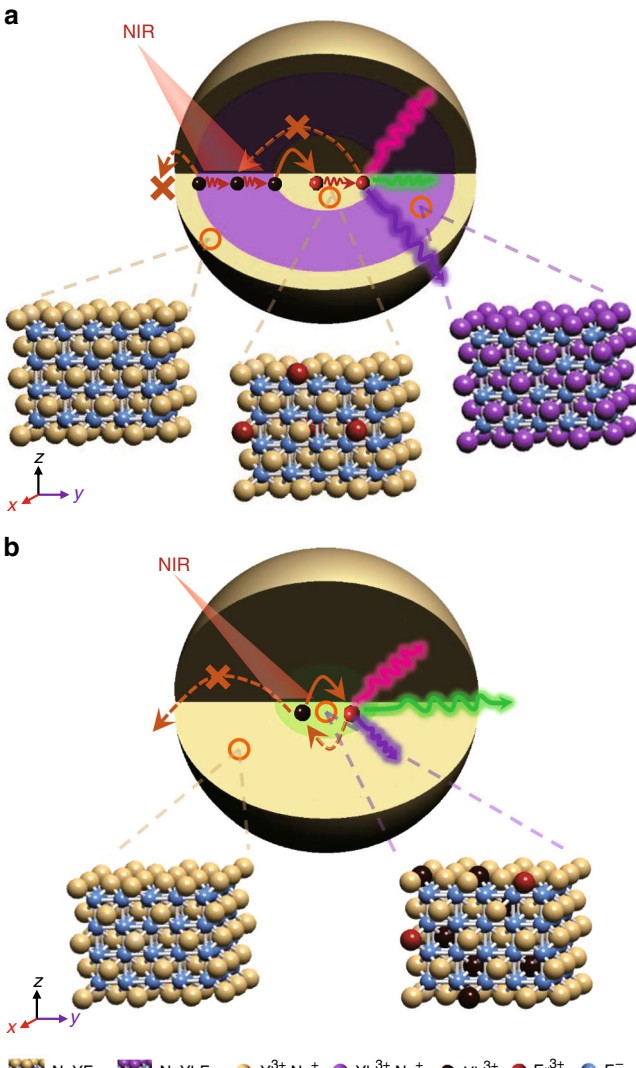

**Fig. 1 Schematic diagram for the UCNPs showing principle emissions. a** α-NaYF$_4$:Er@NaYbF$_4$@NaYF$_4$ core-shell-shell nanoparticles. **b** α-NaYF$_4$:20% Yb, 2% Er@NaYF$_4$ core-shell nanoparticles. Both backward energy transfer and surface quenching processes can be efficiently suppressed in **a**.

nanoparticles was confirmed by X-ray diffraction and transmission electron microscopy (Fig. 2a–c and Supplementary Fig. 3). Typically, the as-synthesized α-NaY$_{0.78}$F$_4$:Yb$_{0.2}$Er$_{0.02}$@NaYF$_4$ nanoparticles, which are one of the most efficient ensembles for photon upconversion[24], generate intense green emission ($^2$H$_{11/2}$, $^4$S$_{3/2}$ → $^4$I$_{15/2}$), competitive red emission ($^4$F$_{9/2}$ → $^4$I$_{15/2}$) and weak violet emission ($^2$H$_{9/2}$ → $^4$I$_{15/2}$) for Er$^{3+}$ under a 980-nm excitation (Fig. 2e, Supplementary Fig. 4); these findings are consistent with previous reports[24,25]. The inert shell herein (~10.5 nm, Supplementary Fig. 3) is thick enough to suppress surface or solvent quenching of the UCL[26,27]. In stark contrast, once the activator and sensitizer were separately doped into the core and intermediate layer of the core-shell-shell nanoparticles at a higher doping level, these α-NaYF$_4$:10% Er@NaYbF$_4$ @NaYF$_4$ (C$_Y$:10% Er@S$_{Yb}$@S$_Y$) nanoparticles exhibited a markedly increased UC emission, with an ~100-fold increment, especially for the multiphoton violet emission at ~407 nm (Fig. 2e, f). Both the violet and red emission are predominant in this multilayered structure, unlike the strong green emission in the canonical structure (Fig. 2g). Moreover, all of the multilayered structures with Yb$^{3+}$

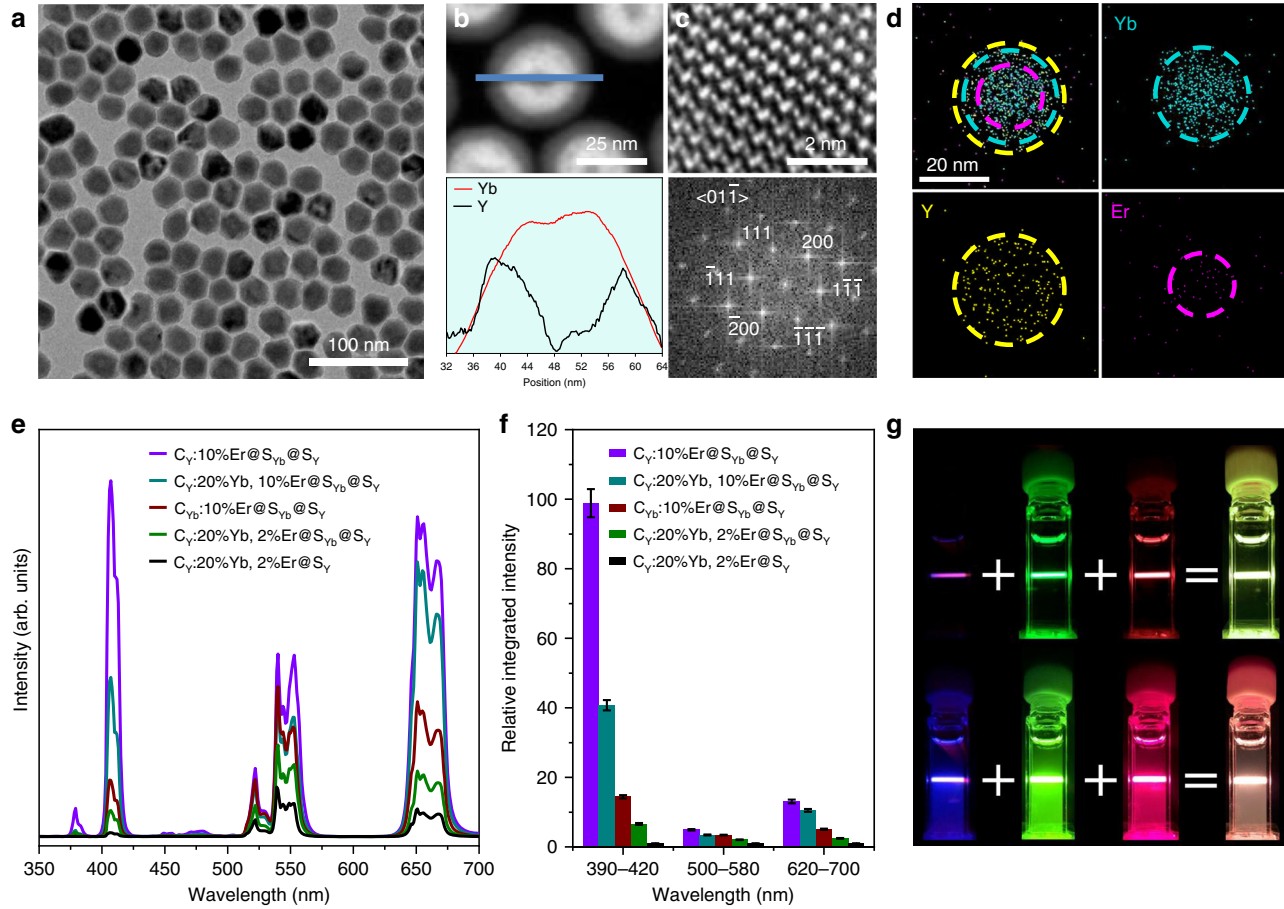

**Fig. 2 Efficient multiphoton upconversion in trilayered nanoparticles. a–d** TEM (**a**) and high-resolution STEM images (**b**, **c**) revealing the single-crystalline nature of the α-NaYF$_4$:Er@NaYbF$_4$@NaYF$_4$ nanocrystal ($d_{SYb}$ = 8.3 ± 0.7 nm, $d_{SY}$ = 2.3 ± 0.1 nm). The $d_{SYb}$ and $d_{SY}$ are used to designate the thickness of the intermediate NaYbF$_4$ and the outmost NaYF$_4$ shells, respectively. The EDX results (**b**, **d**) show distribution of Er$^{3+}$, Yb$^{3+}$ and Y$^{3+}$ ions, consistent with the core-shell-shell structure. Nanoparticles with $d_{SY}$ = 5.6 ± 0.7 nm were selected for better contrast in **b**. **e** Upconversion emission spectra (collected with the mode of "emission correction off") for different UCNPs upon 980-nm excitation (24.0 W cm$^{-2}$): C$_Y$, C$_{Yb}$, S$_{Yb}$ and S$_Y$ are used to designate NaYF$_4$ core, NaYbF$_4$ core, NaYbF$_4$ shell and NaYF$_4$ shell, respectively. For C$_Y$:20%Yb, 2%Er@S$_Y$, $d_{SY}$ = 10.5 ± 0.8 nm. **f**, **g** The PL intensities (**f**) and luminescent photographs (**g**) for C$_Y$:20% Yb, 2% Er@S$_Y$ (top) and C$_Y$:10% Er@S$_{Yb}$@S$_Y$ nanoparticles (bottom). Error bars in **f** represent the standard deviation of three trials. Source data are provided as a Source Data file. The photographs were taken through violet, green and red color filters upon 980-nm excitation (15.2 W cm$^{-2}$).

and Er$^{3+}$ codoped into the core present a much lower UC luminescent intensity than our C$_Y$:10% Er@S$_{Yb}$@S$_Y$ structure despite their particle sizes, sensitizer layers and inert layers all being nearly the same. This dynamic reveals that a much more efficient energy transfer upconversion occurs in this core-shell-shell structure, indicating the superiority of combining energy migration and interfacial energy transfer in the upconversion process. It should be pointed out that severe luminescence quenching will occur if the sensitizing layer is not encapsulated by an inert shell in this structure (Supplementary Fig. 5), indicating that the surface quenching is minimized by this means. It is also noted that the green emission in the C$_Y$:10% Er@S$_{Yb}$@S$_Y$ structure increases by only fivefold compared with the canonical structure, consistent with the Er$^{3+}$ content increasing from 2 to 10%. This finding may indicate population saturation for the $^2$H$_{11/2}$ and $^4$S$_{3/2}$ levels of Er$^{3+}$, unlike those emitting violet and red light.

Enrichment of excitation energy can be realized not only by increasing the Yb$^{3+}$ doping level to 100% but also via thickening of the sensitizing layer ($d_{SYb}$). The UC emission for the C$_Y$:10% Er@S$_Y$:Yb@S$_Y$ UCNPs increases continuously with Yb$^{3+}$ concentration in the sensitizing layer, with no luminescence

quenching observed (Supplementary Fig. 6). This phenomenon demonstrates that the contribution of cross-relaxation or self-quenching among Yb$^{3+}$ ions to concentration quenching can be neglected, consistent with a previous report[16]. With increasing $d_{SYb}$ (Supplementary Fig. 7 and Supplementary Table 1), the overall UC emission and QY for the UCNPs also enhances continuously until $d_{SYb}$ = 8.3 nm (Fig. 3a, b). In particular, the violet and red emission increases much faster than the green emission, resulting in a constantly increased contribution from the violet and red emission to the overall emission. While the thickness of NaYbF$_4$ shell further increases, the overall UC emission and QY for the UCNPs decreases instead. The QY of the optimum UCNPs firstly increases with increasing excitation power and then reaches saturation when the excitation power is relatively higher. Once $d_{SY}$ is thicker than 2 nm, the inert shell thickness ($d_{SY}$, determined by the transmission electron microscopy (TEM) images in Supplementary Fig. 8 with details shown in Supplementary Table 1) seems to have little effect on the population of different Er$^{3+}$ energy levels or the overall intensity of the UC emission (Fig. 3c, d). This dynamic is due to the luminescence from Yb$^{3+}$ being more likely to be quenched by surface defects instead of solvent quenching and due to the

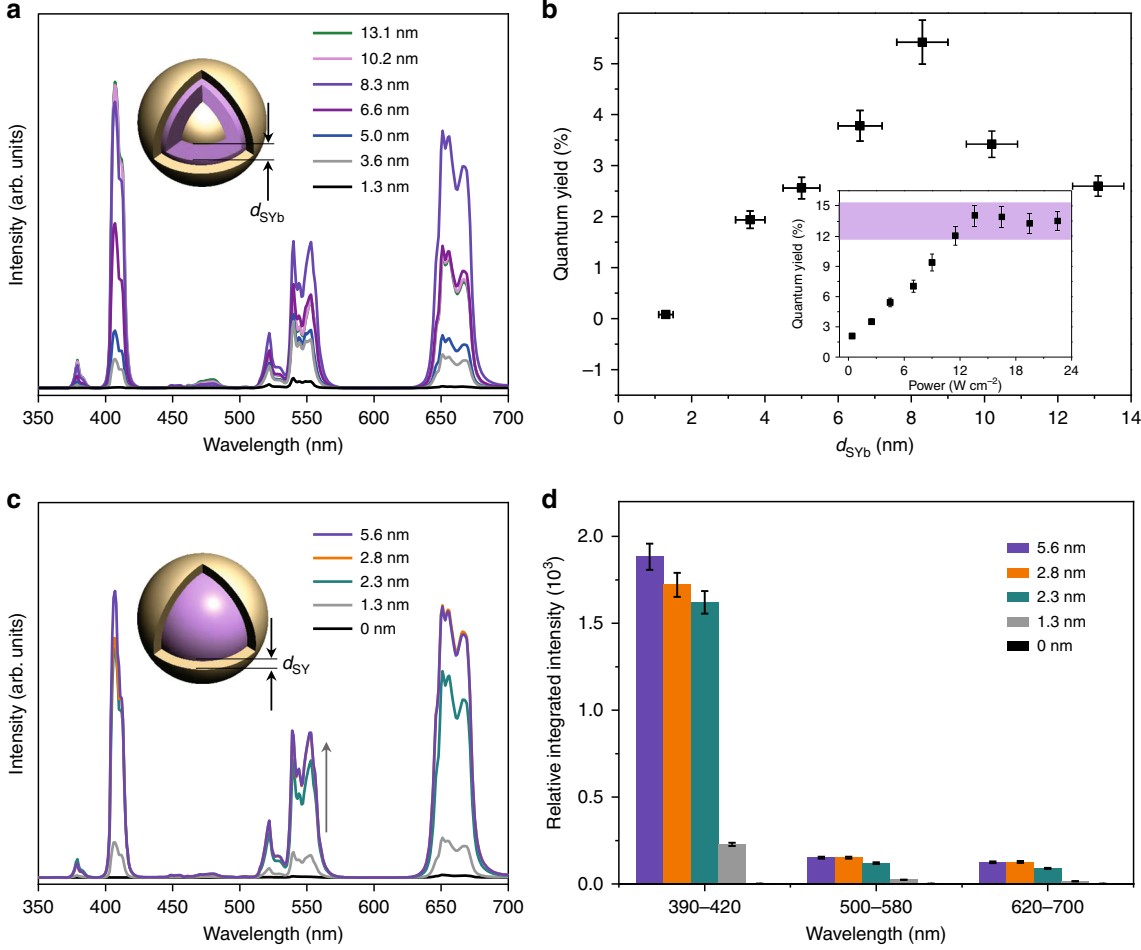

**Fig. 3 Shell thickness-dependent upconversion luminescence. a, c** Upconversion emission spectra for the $C_Y$:10% Er@$S_{Yb}$@$S_Y$ core-shell-shell nanoparticles with varying shell thickness: **a** NaYbF$_4$ shell, $d_{SY}$ = 2.0–2.7 nm; **c** NaYF$_4$ inert shell, $d_{SYb}$ = 8.3 ± 0.7 nm (collected with the mode of "emission correction off"). **b** The quantum yields (QY) as a function of NaYbF$_4$ shell thickness (4.5 W cm$^{-2}$). The inset: power-dependent quantum yields for NPs with $d_{SYb}$ of 8.3 ± 0.7 nm and $d_{SY}$ of 2.3 ± 0.1 nm. The error bars for QY represent the standard deviation of three trials. Source data are provided as a Source Data file. The violet shading in the inset is used to mark saturated QY. **d** Corresponding emission intensity for violet, green and red bands when the thickness of NaYF$_4$ shell varies. The results indicate the key roles of excitation energy enrichment and inert shell protection in enhancement of multiphoton upconversion for the core–shell–shell nanoparticles. The standard deviation of three trials are shown as error bars. Source data are provided as a Source Data file.

relatively thin inert shell being sufficient to suppress the quenching[26]. In the meantime, the distance of the Er$^{3+}$ from the surface is larger than 10 nm ($d_{SYb} + d_{SY}$) in this case, which is sufficient to prevent the solvent quenching of Er$^{3+}$ emission[26,27].

**Evidence of an efficient interfacial energy transfer**. The power dependence for the UC emission from the $C_Y$:10% Er@$S_{Yb}$@$S_Y$ multilayered nanoparticles (Supplementary Fig. 9) indicates that the 407-nm emission arises from the three-photon UC process, while the 540-nm emission is governed by the two-photon UC process. The slop factor calculated for the red emission is 2.3, suggesting contributions from both two- and three-photon processes (Fig. 4a)[23,27–29]. As we discussed above, the population of the green emitting levels for Er$^{3+}$ may be saturated because two-photon UC is more efficient under low excitation power density[19,30]. On average, the amount of Yb$^{3+}$ ions increases by ~22-fold when $d_{SYb}$ varies from 1.3 to 8.3 nm, with the green emission intensity increasing by ~20-fold correspondingly (Supplementary Fig. 10). This finding indicates that the green emission intensity increases linearly with the increment of Yb$^{3+}$ sensitizer ions. In contrast, the violet and red emission intensities increase by 440 and 180 times, respectively, in the same

condition, further validating the hypothesis that the $^4F_{9/2}$ level may be populated primarily through a triphotonic transition accompanied by back energy transfer (Fig. 4a). The three-photonic population of the $^2H_{9/2}$ and the $^4F_{9/2}$ levels is greatly enhanced; thus, the violet and red emissions increase exponentially when $d_{SYb}$ is elevated. This finding explains why the contributions of the violet and red emissions for $C_Y$:10% Er@$S_{Yb}$@$S_Y$ UCNPs become dominant at larger $d_{SYb}$. Whereas once $d_{SYb}$ is larger than ~10.2 nm, the violet emission gradually reaches saturation while both the green and red emissions obviously decay. This phenomenon implies that the population of the $^4F_{9/2}$ level through a triphotonic transition may be hindered when a much thicker NaYbF$_4$ shell is coated.

To shed more light on the energy transfer between Yb$^{3+}$ and Er$^{3+}$ ions in the multilayered $C_Y$:10% Er@$S_{Yb}$@$S_Y$ UCNPs, the decay kinetics of the Er$^{3+}$ emission bands from these nanoparticles were measured and compared with those obtained from the canonical $C_Y$:20% Yb, 2% Er@$S_Y$ samples. In comparison with the canonical structure, the separated distribution of Yb$^{3+}$ and Er$^{3+}$ ions in $C_Y$:10% Er@$S_{Yb}$@$S_Y$ UCNPs undoubtedly slows the energy transfer from Yb$^{3+}$ to Er$^{3+}$ and prolongs the decay time for the Yb$^{3+}$ $^2F_{5/2}$ level (Fig. 4b). Consequently, the rise and decay

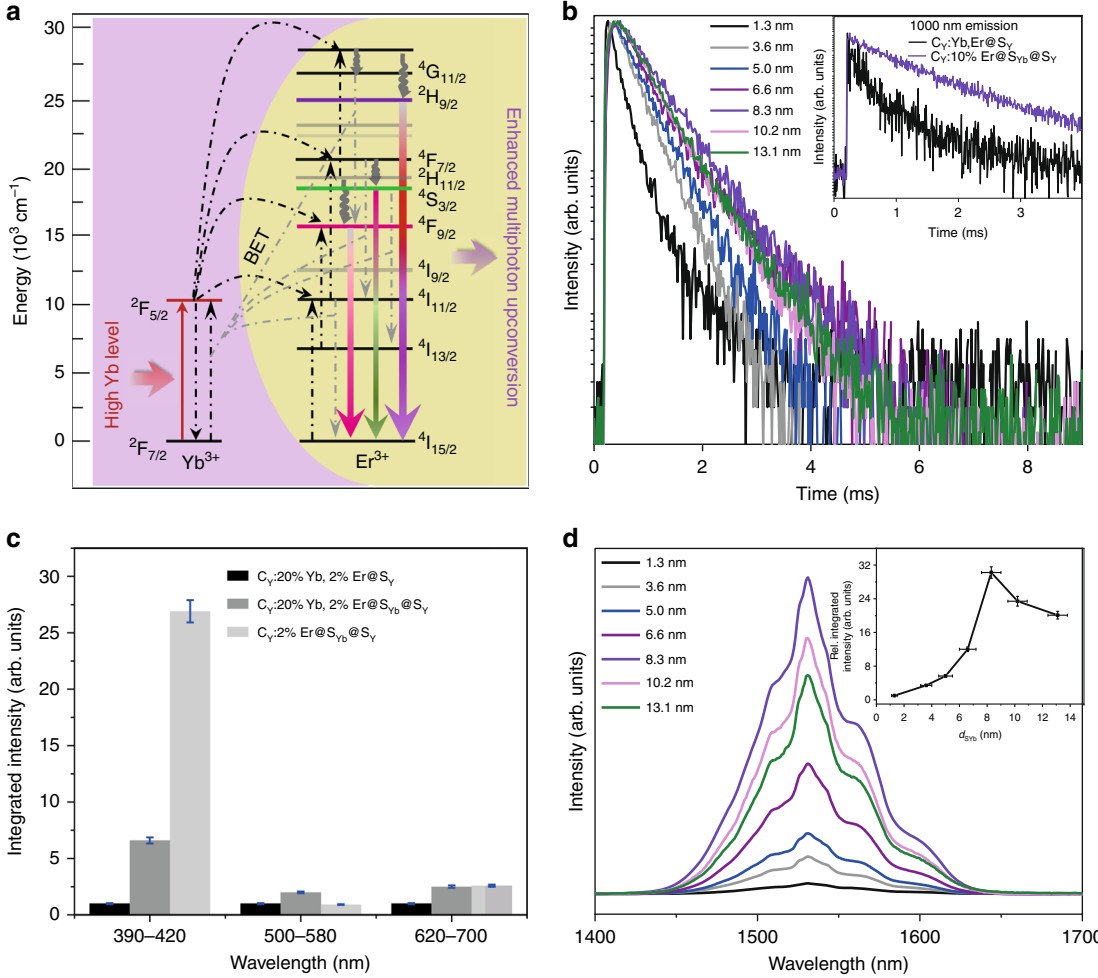

**Fig. 4 Efficient interfacial energy transfer in trilayered nanoparticles. a** Simplified energy-level diagram shows enhanced multiphoton upconversion from $Yb^{3+}$ to $Er^{3+}$ in the $C_Y$:10% $Er@S_{Yb}@S_Y$ core-shell-shell nanoparticles ($d_{SYb} \geq 3.2$ nm, $d_{SY} \geq 2.0$ nm). **b** Luminescence decay curves of $Er^{3+}$ emission at 407 nm for $C_Y$:10% $Er@S_{Yb}@S_Y$ nanoparticles with $d_{SYb}$ increasing from 1.3 to 13.1 nm ($d_{SY} = 2.0$–2.7 nm). Inset shows the decay curves of $Yb^{3+}$ emission for the as-synthesized $C_Y$:10% $Er@S_{Yb}@S_Y$ core-shell-shell ($d_{SYb} = 8.3 \pm 0.7$ nm) and $C_Y$:20% Yb, 2% $Er@S_Y$ core-shell nanoparticles ($d_{SY} = 10.5 \pm$ 0.8 nm). **c** Relative integrated UCL intensity of 2% $Er^{3+}$-doped UCNPs (for trilayered structure, $d_{SYb} = 8.3 \pm 0.7$ nm, $d_{SY} = 2.3 \pm 0.1$ nm; for core-shell structure, $d_{SY} = 10.5 \pm 0.8$ nm). **d** NIR downshifting luminescence spectra and the relative integrated intensities (inset) for the $C_Y$:10% $Er@S_{Yb}@S_Y$ nanoparticles with varying $NaYbF_4$ shell thickness ($d_{SY} = 2.0$–2.7 nm). The error bars in (**c**) and (**d**) (inset) represent the standard deviation of three trials. Source data are provided as a Source Data file.

times for the $Er^{3+}$ emission are obviously prolonged in this core-shell-shell structure (Supplementary Fig. 11) since $Yb^{3+} \rightarrow Er^{3+}$ energy transfer occurs at the interface following $Yb^{3+}$-$Yb^{3+}$ energy migration in the sensitizer layer[25]. Moreover, with the increment of the sensitizers ($d_{SYb}$), all the rise and decay times for the upconverted $Er^{3+}$ emission at 407, 540, and 651 nm increase constantly until $d_{SYb} = 8.3$ nm (Supplementary Fig. 12). This finding supports our conjecture that the excitation energy absorbed by the outmost $Yb^{3+}$ ions can migrate a longer distance through the $NaYbF_4$ layer and transfer to the $Er^{3+}$ levels at the core-shell interface. However, luminescent quenching occurs if $NaYbF_4$ layer is too thicker (larger than 10.2 nm), as demonstrated by the shortened lifetimes of the emission bands. It is also important to note that the UCL at 407 nm has single exponential decay kinetics for all the samples, except for the one with the thinnest sensitizing layer (~1.3 nm). In addition to the long decaying component resembling that of the others, thin protection layer (<4.0 nm) may lead to fast decay of $^2H_{9/2}$ state from $Er^{3+}$ ions neighboring sensitizing layer. In the meantime, the decay time for the red emission at 651 nm increases more strongly. This may be due to the increased contribution of the

three-photon process with increasing amount of added $NaYbF_4$. Similar to their UC emission intensities, the lifetimes of the emission bands at 407, 540 and 651 nm initially increase until $d_{SY} = 2.3$ nm, and then, they reach an approximately constant maximum value (Supplementary Fig. 13), thereby verifying that a thin inert layer is enough for suppression of surface quenching in this structure.

To preclude the possibility that higher activator concentration (10 vs. 2%) plays a key role in enhancing energy transfer efficiency, we also synthesized $C_Y$:2% $Er@S_{Yb}@S_Y$ UCNPs for comparison. The UCL intensity for the three 2% Er-doped UCNPs in Fig. 4c provides clear evidence for highly efficient interfacial energy transfer. The three-photo upconversion is enhanced by 6.6-fold if we coat an $NaYbF_4$ layer onto the canonical core, while a 27-fold enhancement can be achieved if $Yb^{3+}$ and $Er^{3+}$ ions are separately located despite the total amount of $Yb^{3+}$ ions being lower in the latter than that in the former. This finding also indicates that the cross-relaxation between the sensitizer $Yb^{3+}$ and the activator $Er^{3+}$ may contribute more to energy loss than that among the sensitizer $Yb^{3+}$ ions. The fact that insertion of an inert layer between activator core and sensitizer layer will decrease UCL intensity of the

multilayered UCNPs (Supplementary Fig. 14) also confirms a higher efficient energy transfer at activator@sensitizer interface. The near infrared (NIR) downshifting luminescence spectra in Fig. 4d further validate such a highly efficient energy transfer in this structure. As discussed above, the amount of $Yb^{3+}$ ions increases by ~22-fold when $d_{SYb}$ varies from 1.3 to 8.3 nm, with the NIR emission intensity increasing by ~30-fold correspondingly. This phenomenon validates the fact that both upconversion and downconversion efficiencies increase when $Yb^{3+}$ and $Er^{3+}$ ions are spatially separated. Whereas the NIR emission will decay if $NaYbF_4$ layer is further thickened, similar to that observed in UC emission. Furthermore, the increase in concentration of $Er^{3+}$ from 10 to 50% in the core results in a slight increase in the overall upconversion emission (Supplementary Fig. 15), different from the recently reported nanostructures (Supplementary Fig. 16)[16,18]. Obvious concentration quenching occurs only after the concentration of $Er^{3+}$ reaches 70%, suggesting successful suppression of concentration quenching. As anticipated, the energy transfer from $Yb^{3+}$ to $Er^{3+}$ accelerates with increasing $Er^{3+}$ concentration (Supplementary Fig. 17). Surprisingly, the lifetimes for the upconverted $Er^{3+}$ emission bands increase with increasing $Er^{3+}$ concentration from 2 to 30% (Supplementary Fig. 18), unlike the lifetime shortening observed with increasing $Er^{3+}$ content from 2 to 16% in the $NaYF_4@NaYbF_4:Er@NaYF_4$ UCNPs[16]. The prolonged rise time for the $Er^{3+}$ emission in 10 and 30% $Er^{3+}$-doped samples may hinder $Er^{3+}$-$Er^{3+}$ interactions that produce nonradiative decay pathways, leading to fast decay of $^2H_{9/2}$ state (Supplementary Table 2). This phenomenon further validates the fact that interfacial energy transfer enables an effective upconversion process with suppressed concentration quenching. In the meantime, the luminescent property of the larger $C_Y$:10% $Er@S_{Yb}@S_Y$ UCNPs with a similar $d_{SYb}$ resembles that of the smaller samples (Fig. 3a and Supplementary Fig. 19), thus confirming the key role of $d_{SYb}$ in UCL for this multilayered structure.

The temperature-dependent photoluminescent (PL) properties of $Er^{3+}$ in the trilayered $C_Y$:10% $Er@S_{Yb}@S_Y$ UCNPs give further evidence for suppression of $Yb^{3+}$-$Er^{3+}$ cross-relaxation. Obviously, the violet emission band is greatly intensified at cryogenic temperatures, with an ~20-fold enhancement, whereas the red emission shows only an ~3.5-fold increase (Supplementary Figs. 20, 21), resulting in a dominant violet emission from the UCNPs. As a consequence, the $C_Y$:10% $Er@S_{Yb}@S_Y$ UCNPs show temperature-dependent luminescence colors ranging from orange-red to violet when the temperature is decreased from 298 to 3.8 K. In contrast, the red emission increases by ~22-fold for the canonical $NaYF_4$:20% Yb, 2% $Er^{3+}@ NaYF_4$ nanocrystals (Supplementary Fig. 22) at cryogenic temperatures, similar to that observed in a previous report[31]. Avoiding surface quenching by using a thick inert shell may result in higher temperature sensitivity for both UNCPs compared to previous reports[32–34]. It is reasonable that multiphonon-assisted relaxation will be greatly suppressed when temperature is decreased (Supplementary Fig. 23); therefore, three-photon upconversion will be enhanced in both nanostructures[31]. The cross-relaxation between $Er^{3+}$ and $Yb^{3+}$ ions in the canonical structure is favorable for the population of the $Er^{3+}$ $^4F_{9/2}$ level following the three-photon upconversion process. The segregation of $Er^{3+}$ and $Yb^{3+}$ ions in the core-shell-shell structure significantly suppresses the $Er^{3+}$-$Yb^{3+}$ cross-relaxation, thus enabling an increase in the population of the $Er^{3+}$ $^2H_{9/2}$ through the three-photon upconversion process.

**Multiphoton upconversion enhancement in various structures.**
The construction strategy for such a sensitizer-activator separated multilayered core-shell-shell structure is an effective and general strategy for selective enhancement of multiphoton upconversion emission from different activators in various hosts (Supplementary Figs. 24–29). In comparison with the canonical nanoparticles, the UCL intensity for $\beta$-$NaYF_4$:$Er@NaYbF_4@NaYF_4$ and $LiYF_4$:$Er@LiYbF_4@LiYF_4$ UCNPs, especially their intensity for violet and red emission, is significantly increased (Supplementary Figs. 24, 25), consistent with that observed in $\alpha$-$NaYF_4$. Similarly, the UC emission bands attributed to the transition from the higher energy levels of $Tm^{3+}$ ($^1D_2$) and $Tb^{3+}$ ($^5D_3$) are remarkably intensified in these $\alpha$-$NaYF_4$:$A@NaYbF_4@NaYF_4$ UCNPs (Supplementary Figs. 26–29). In the canonical $\alpha$-$NaYF_4$:$Yb/Tm@NaYF_4$ nanoparticles, the UCL intensity of $Tm^{3+}$ in the range of 300–600 nm gradually decreases when the concentration is higher than 0.5 mol% (Supplementary Fig. 27), indicating characteristic concentration quenching[19]. Conversely, the integrated UCL intensity of $Tm^{3+}$ (300–600 nm) in our trilayered structures continuously increases with increasing concentration of $Tm^{3+}$ to 8 mol%, although the emission band at ~800 nm exhibits a complex variation trend. In addition, substantial luminescence reduction can also be observed when 10 mol% $Tm^{3+}$ is doped into the core. The emissive enhancement of $Tb^{3+}$ in core-shell-shell nanoparticles is also remarkable if we compare the two luminescent photographs (Supplementary Fig. 28). The core-shell UCNPs with $Yb^{3+}$/$Tb^{3+}$ codoped into the core emit weak green light, while core-shell-shell UCNPs emit bright blue light. All radiative transitions from $^5D_3$ to $^7F_{2,4,5,6}$ are significantly enhanced and predominate (Supplementary Fig. 29), suggesting an increased population of $^5D_3$.

The highly efficient upconversion energy transfer in these trilayered core-shell-shell nanoparticles was also verified by QY measurements. Taking the UC emission of $Er^{3+}$ as an example, the canonical nanoparticles with a thick inert shell present a QY of 0.61% at an excitation power density of 4.5 W cm$^{-2}$, while all three trilayered UCNPs prepared in this work exhibit a QY of 5–6% despite having different sensitizer layer thicknesses and different lattice hosts (Table 1). These values are higher than most of the reported values measured at relatively higher excitation power density (Supplementary Table 3, Supplementary Fig. 30).

This concentration quenching suppression strategy enables versatile design of multilayered structures based on lanthanide-doped nanoparticles as bright and multicolor phosphors. As a proof-of-concept experiment, we synthesized four-layered $\alpha$-$NaYF_4$:$A_1@NaYbF_4@NaYF_4$:$A_2@NaYF_4$ UCNPs with dual activators (Supplementary Table 4). Spatial separation of $Yb^{3+}$, $Er^{3+}$, and $Tm^{3+}$ enables $\alpha$-$NaYF_4$:$Tm@NaYbF_4@NaYF_4$: $Er@NaYF_4$ nanoparticles to emit bright light that appears cool white to the naked eye (Supplementary Fig. 31). The emission bands consist of those from both $Er^{3+}$ and $Tm^{3+}$ without deleterious cross-relaxation and cover the whole ultraviolet-visible range. In contrast, weaker and fewer emission bands are observed in $\alpha$-$NaYF_4$:$Yb/Tm/Er@NaYF_4$ nanoparticles at the same doping level,

---

**Table 1 Upconversion quantum yields of $Er^{3+}$-doped UCNPs synthesized in this work.**

| Sample | Size (nm) | QY (%)[a] |
|---|---|---|
| $\alpha$-$NaYF_4$:20%Yb,2%Er@$NaYF_4$ | 32.0 ± 2.5 | 0.61 ± 0.20 |
| $\alpha$-$NaYF_4$:10%Er@$NaYbF_4$@$NaYF_4$ ($d_{SYb}$ = 8.3 ± 0.7 nm, $d_{SY}$ = 2.3 ± 0.1 nm) | 32.2 ± 2.2 | 5.42 ± 0.43 |
| $\alpha$-$NaYF_4$:10%Er@$NaYbF_4$@$NaYF_4$ ($d_{SYb}$ = 8.3 ± 0.7 nm $d_{SY}$ = 5.6 ± 0.7 nm) | 38.8 ± 3.8 | 6.34 ± 0.48 |
| $\beta$-$NaYF_4$:10%Er@$NaYbF_4$@$NaYF_4$ | 31.9 ± 1.3 | 6.82 ± 0.50 |

[a]Excitation power density: 4.5 W cm$^{-2}$, QYs are means ± s.d. of three trials. Source data are provided as a Source Data file.

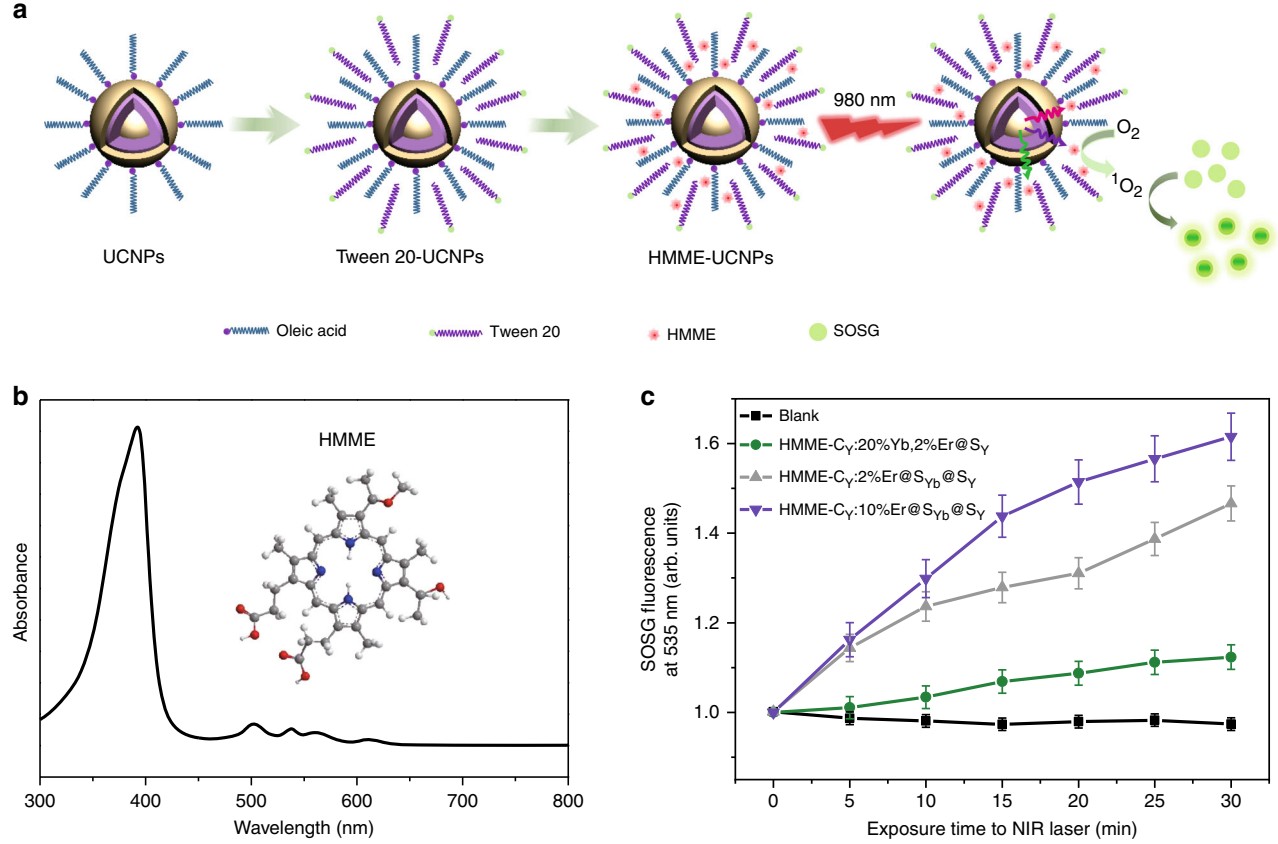

**Fig. 5 Singlet oxygen production activity upon exciting the UCNPs. a** Schematic diagram for loading hematoporphyrin monomethyl ether (HMME) and producing $^1O_2$ on the UCNPs. **b** Structural formula and absorption spectrum of HMME. **c** Comparison of $^1O_2$ production under 980 nm irradiation (24 W cm$^{-2}$) as determined by the augmentation of singlet oxygen sensor green reagent (SOSG) fluorescence at 535 nm. For $C_Y$:20%Yb, 2%Er@$S_Y$, $d_{SY}$ = 10.5 ± 0.8 nm. For $C_Y$:Er@$S_{Yb}$@$S_Y$, $d_{SYb}$ = 8.3 ± 0.7 nm, $d_{SY}$ = 2.3 ± 0.1 nm. The error bars represent the standard deviation of three trials. Source data are provided as a Source Data file.

which reveals substantial quenching and cross-relaxation between $Er^{3+}$ and $Tm^{3+}$. Similarly, intensified multiphoton emission from both $Tm^{3+}$ and $Tb^{3+}$ can be found in four-layered UCNPs doped with dual activators (Supplementary Fig. 32). The sensitizing layer here not only favors enriching excitation energy but also prohibits cross-relaxation between different activators, which assures bright and versatile upconversion emission from these multilayered UCNPs. This dynamic will benefit the design and construction of highly efficient UCL nanomaterials for various applications[2,6,30,35,36].

Based on the above results, we hypothesize that separation location for the $Yb^{3+}$ ions and $Er^{3+}$ ions may not only alleviate $Yb^{3+}$-$Er^{3+}$ ion cross-relaxation, but also hinder nonradiative decay pathways induced by the cross-relaxation between $Er^{3+}$ ions through prolonging the population process of the emitting levels. Consequently, concentration quenching will be suppressed in certain degree, leading to bright upconversion emission, especially stronger emission at shorter wavelengths, from these core-shell-shell nanoparticles. Although the energy transfer upconversion in $Tm^{3+}$- and $Tb^{3+}$-doped UCNPs is much more complex and needs further investigation, enhanced multiphoton upconversion can also be realized in this structure.

The intensified emission from high-energy levels may offer more opportunities for deep-tissue biophotonics, such as photodynamic therapy and optogenetics. Singlet oxygen ($^1O_2$) production activity of hematoporphyrin monomethyl ether (HMME), a commercial photosensitizer with strong absorption around 400 nm, upon exciting the UCNPs was tested here to validate our conjecture (Fig. 5a, b, Supplementary Fig. 33). Due to the weak absorption of HMME in the range of 500–650 nm, the augmentation of singlet oxygen sensor green reagent (SOSG) fluorescence in aqueous dispersion of $C_Y$:20% Yb, 2% Er@$S_Y$ UCNPs is about 12.3% after 980 nm laser irradiation for about 30 min (Fig. 5c and Supplementary Fig. 34). In contrast, the SOSG fluorescence augmentation is about 46.4% and 61.5%, respectively, in aqueous dispersion of $C_Y$:2% Er@$S_{Yb}$@$S_Y$ and $C_Y$:10% Er@$S_{Yb}$@$S_Y$ UCNPs. While the fluorescence of the control sample decreases about 2.8%. As compared with $C_Y$:20% Yb, 2% Er@$S_Y$ UCNPs, neglected enhancement in green and red emissions for $C_Y$:2% Er@$S_{Yb}$@$S_Y$ UCNPs was observed (Fig. 4c); thus, the improved $^1O_2$ production activity for the latter two samples should be attributed to their strong violet emission. The invariable lifetimes (Supplementary Fig. 35) for upconverted $Er^{3+}$ emission before and after loading HMME exclude Förster resonance energy transfer (FRET) mechanism. In general, only a $^1O_2$ QY of a photosensitizer can be detected by using Rose Bengal (RB) with a high $^1O_2$ QY of 0.76 as a model photosensitizer[37]. Even though $^1O_2$ QY of our UCNPs cannot be detected since these UCNPs are excited by infrared light, different from 488 nm for RB excitation. These results demonstrate $C_Y$:10% Er@$S_{Yb}$@$S_Y$ UCNPs do emit much stronger violet light and initiate photoreactions more efficiently than the canonical UCNPs, indicating that they are superior to the latter in potential applications such as optogenetics.

## Discussion

We have demonstrated the critical role of cross-relaxation between sensitizers and activators in concentration quenching in lanthanide-doped UCNPs by designing and synthesizing different structures in which the codopants are distributed differently. We have revealed that spatially separated distribution of sensitizers and activators in neighboring layers can suppress their cross-relaxation and enable highly efficient energy transfer upconversion at the interface. As a result, the UCL from a multilayered structure $NaYF_4:Er@NaYbF_4@NaYF_4$ is significantly enhanced at relatively high doping levels, especially for multiphoton upconversion emission. Therefore, a high QY of 6.34% is reached in optimum $NaYF_4:Er@NaYbF_4@NaYF_4$ nanoparticles at a low excitation power density of 4.5 W cm$^{-2}$. Moreover, the suppression of sensitizer-activator cross-relaxation enables the further enhancement of multiphoton upconversion emission at cryogenic temperatures. This segregated doping strategy provides an easy and efficient way to design and synthesize bright, versatile lanthanide-doped UCNPs to meet the demands of various applications.

## Methods

**Nanocrystal synthesis**. We synthesized the $NaYF_4:Yb,Er@NaYF_4$ and $NaYF_4:Er@NaYbF_4@NaYF_4$ nanoparticles following a procedure reported by Mai et al. with a slight modification[24]. Detailed experimental procedures are shown in Supplementary Methods.

**Nanocrystal characterization**. The phase and crystal structure of the samples were characterized by powder X-ray diffractometer (PANalytical X'pert PRO-DY2198). The size, shape and element distribution of the samples were observed by TEM (Jeol JEM 2100F) combined with energy dispersive X-ray spectrum operating at an acceleration voltage of 200 kV. Steady-state PL measurements were performed on an Edinburgh FLS 980 spectrofluorometer in conjunction with a continuous-wave (CW) 980-nm diode laser at an excitation power density of 24.0 W cm$^{-2}$. Samples for temperature-dependent PL measurements were prepared using a drop-casting method on a quartz glass substrate. An Oxford Optistat DryBL4 cryostat and a Microstat HiRes2 with a temperature controller (Mercur-yiTC) were used for lowering (3.8–298 K) and mounting temperature (298–508 K), respectively. The samples were held at a certain temperature for at least 10 min to assure equilibration. Time-resolved PL spectra were collected by using an optical parametric oscillator (OPO) as the excitation source (197–2750 nm, 20 Hz repetition rate and ~3 ns pulse width). An excitation power density of 4.5 W cm$^{-2}$ was used for all the measurements except for power-dependent QY measurements for $C_Y$: 10%Er@$S_{Yb}$@$S_Y$ NPs ($d_{SYb} = 8.3 \pm 0.7$ nm, $d_{SY} = 2.3 \pm 0.1$ nm).

**Detection of $^1O_2$ production**. The $NaYF_4:Yb,Er@NaYF_4$ and $NaYF_4:Er@NaYbF_4@NaYF_4$ nanoparticles were loaded with HMME after modification by Tween 20, and then suspended in 2 mL of a SOSG aqueous solution. The mixture was injected into a quartz cuvette placed on a magnetic stirring apparatus and the solution was irradiated with an 980 nm laser at 24 W cm$^{-2}$ for 5 min time intervals beginning from time ($t$) = 0–30 min, with the fluorescence emission of SOSG (excited by 380 nm) being measured between intervals using FLS980.

## Data availability

The data that support the findings of this study are available from the corresponding author upon reasonable request. The source data underlying Figs. 1a, 2f, 3b, d, 4c, d, 5c and Table 1 are provided as a Source Data file.

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

## Acknowledgements

The National Natural Science Foundation of China (No. 21673090), the National Key Research and Development Program of "Strategic Advanced Electronic Materials" (2016YFB0401100), Hubei Provincial Natural Science Foundation of China (2019CFA002), and the Fundamental Research Funds for the Central University (2019kfyXMBZ018) supported this work. Here, we thank Prof. X.Y. Chen (Fujian Institute of Research on the Structure of Matter, Chinese Academy of Sciences) for helpful discussion.

## Author contributions

The scientific concepts, ideas and experimental designs were the result of interactions and discussions between B.Z., Y.M., Z.G., J.Y., T.Z. B.Z. and C.Q. synthesized the nanoparticles and conducted the electron microscopy. B.Z. and B.T. conducted the spectroscopic measurements and the QY measurements. B.T. detected $^1O_2$ production. B.Z., Y.M. and C.Z. wrote the paper, in coordination with all the authors.

## Competing interests

The authors declare no competing interests.
