## [Peer Review File · Nature Communications]

Reviewers' comments:

Reviewer #1 (Remarks to the Author):

The article by Ying Ma and colleagues entitled "Enhancing multiphoton upconversion through interfacial energy transfer in multilayered nanoparticles" presents interesting observation of enhanced upconversion (UC), as soon as the sensitizer ions (Yb) are displaced from activators (Er, and some examples of Tm and Tb are provided as well).

The article is relatively well written, in most cases the graphs are easily readable and clear. The novelty of this idea is deserving presentation, but there are however some shortcomings, which have to be addressed:

1. technical information about measurements in various temperatures is missing

2. measuring QY requires power dependent values and not single points. Moreover, I am also using FLS980 for QY measurements, and it is difficult (if not impossible) to get reliable values for UCQY on this equipment. Did the authors validated their results vs some standard sample and literature data e.g. comparing the obtained results from the publications of group of U.Resch-Genger.

3. it is not clear if the authors used the same concentration of NPs in all experiments to make reliable comparison. There is no technical information on this. How this was achieved ?

4. Discussing the impact core-shell-shell NPs on their spectral properties, more detailed work is necessary to demonstrate the actual composition of dopants is exactly as the authors intended to have. Showing just Fig.1c for single NPs is far too little. Moreover, the 10%Er is big enough to be detected with TEM-EDX - and this must be done to convince readers about reliability of results and materials.

5. There are many samples and tests, but in most cases, there is not enough information linked to the graphs, which precise which sample is used. Just for example - Fig.1 - which sample (in terms of shell thickness) was used here. Such information is missing also for some other graphs and must be corrected / supplemented with proper information.

6. The spectra are not corrected for spectral sensitivity (which includes grating, PMT, optics etc.) - this is not a problem for relative changes, but is crucial if the authors claim the UC emission is X times larger than the green or red one.

7. How the shell thickness was measured and what was the accuracy of this determination ? What are the error bars at the graphs (e.g. Fig.2b and d) - look at my comment on spectral correction as well.

8. Actually, to make the results on Fig.2b meaningful, not (or not only) the intensity versus dSYb should be presented, but the UCQY to account for rising Yb concentration.

9. If only filter was exchanged in front of the camera for pictures 1g, why the positions of the laser beam within the sample is at different heights ? Why there is more scattering in bottom pictures on Fig.1g ? This is also important for UCQY measurements as was shown by Pilch et al. in YbHo UCNPs.

What was the acquisition time for the pictures ? was is the same for all pictures ?

10. Fig.3 - the energy transfer diagram should be confronted with recent papers from M. Bery and E. Chan for the UC and BET. It seems to me not all possible and probable BET mechanisms are shown (e.g. $4I_{11/2} \text{ Er} \rightarrow 2F_{5/2} \text{ Yb}$). Again, which shell thicknesses are presented ? It would be highly interesting to see not only UV/Vis emission but also 1550 nm emission from Er in this new configuration. Panel d does not really mean anything useful for current discussion. How come the low Yb->Er ET (red curve, inset for Fig.3b) makes the UC brighter than the higher Yb->Er ET (corresponding black curve). Yb lifetime should show Yb spontaneous lifetime modified by both outgoing (Yb->Er ET), and incoming (Er->Yb ET) energy ? I understand the lack of Er->Yb BET, but how the energy migrated to Er ions that sit deeper in the NP volume? Is there no CR between Er ions at all ? Very often, the authors of the manuscript describe what they see, but very often I miss deeper explanation, why it happens so, especially, such interesting results are presented. Moreover, the lifetimes (Fig.3b) are counter-intuitive to me: why larger [Yb] concentration (related to larger thickness of dSYb), makes the rise and decays longer ? Energy migration between Yb ions has been observed to apparently increase the lifetime of Yb³⁺, but this was observed in bulk materials (at much longer distances). Moreover, the decays are bi-exponential for me - why? To me, authors should also relate their results to the results of Pilch et al. on the impact of chemical architecture of YbHo on the Ho lifetimes.

11. I don't understand why the authors mismatch alpha and beta phases, NaYF₄ and LiYF₄ hosts - this makes the message more complex, confusing and more difficult to follow. Moreover, either Tb and Tm samples are studied with such care as Er samples (lifetimes, EY etc.), or all this is moved to SI or skipped for clarity. Moreover, Tm is very susceptible to CR both between Tm-Tm pairs and Tm-Yb, why 8% Tm seems to be not susceptible to Tm-Tm CR ? While Yb-Er or Yb-Tm system may indeed show BET from activator to sensitizer and in consequence decrease ETU intensity, in Yb-Tb samples this cooperative process seems to be more complex - although down-shifting effect is known, much more details need to be studied before this Yb-Tb ETU in new "split" system is ready for publication. Moreover, again, the spectral sensitivity correction must be done, otherwise it looks like Tm is emitting in UV only, while in real, 800 nm emission is much stronger than Vis - but current Fig.4c presents uncorrected data and makes the comparison more difficult.

12. Why the lifetimes rise from 2-30% of Er (Fig.S14) for rising [Er] concentration and then fall down ? What are the error bars ? Why there are bi-exponential decays ? What do the rise times tell ?

Concluding - the overall results are very interesting and intriguing, but describing what one can see on the graphs is far too little and in-depth mechanism should be proposed (generally lack of Er->Yb BET is proposed) and additional experiments should be proposed to either support or reject this hypothesis (e.g. power dependent UCQY, TEM-EDX to prove real structure, Yb lifetimes vs rising Er concentration, Er QY vs displacement between Yb and Er shell with yet another empty shell). I know this suggestions are time consuming, but after reading the manuscript I cannot tell I understand what is going on inside. I suppose

modelling such phenomena with rate equations may not be trivial (to account for core-shell), but maybe could help to understand the physics.

Reviewer #2 (Remarks to the Author):

Evaluation:

- Data is technically sound: Data is mostly sound, but there are some holes and missing data that would complete the manuscript.
- Paper provides strong evidence: The manuscript mostly contains strong evidence, but there are some questions and concerns.
- Results are novel: Results are sufficiently novel
- Manuscript is important to scientists in the specific field: Manuscript is sufficiently important to advancing the field.
- Recommendation: The novelty of the results and findings are sufficient and relevant to advancing the field of upconversion nanoparticles for consideration. Considerable for Nature Communications with major revisions addressing the concerns laid out below.

This manuscript reports a two-pronged approach to enhancing upconversion nanoparticle quantum yield, claiming the highest reported quantum yield for these materials at low power density excitations. This is truly a remarkable find, especially if the findings provide such a generalizable strategy as is claimed in the manuscript; however, there are a few concerns regarding some of the background theory, the suggestions implied by the mechanism, some technical details, and some potentially missing data points that would fully round out the manuscript and meet the standards of a Nature Communications publication:

1) The author stated that the rationale, at least partially, for this core-shell-shell structure is that it would alleviate the backward energy transfer pathway in these upconverting nanoparticles systems. Is this pathway significant enough to address directly? It would be appreciated if the authors can supply a stronger rationale and supporting literature to back up this point. Checking citations 11 and 20 as cited by the author, it is not clear that this pathway is so detrimental to the author's goals, and as a matter of fact, it is implied that this back transfer actually assists stronger red and blue emissions, as is shown later in the manuscript.

2) The author mentions a short Yb^{3+} - Er^{3+} distance at the interface, yet later in the manuscript they state that "the rise in the decay times for the Er^{3+} emission are obviously prolonged" due to the same structural feature. This seems inconsistent, and careful wording or explanation would be appreciated.

3) The author states that the canonical structure for UCNPs is a codoped NaYF_4 system, which is true; however, my impression is that the canonical structures are usually β -hexagonal as opposed to α phase used in this paper. It would be therefore important to include the β phase canonical structure in your quantum yield measurements. Furthermore, checking your quantum yield comparisons table (Supplementary Table 1), it is interesting to note that there is a large size discrepancy between the β phased particles and the α phased particles. This is of particular concern considering that within the α phase particles, there is a $\sim 1\%$ difference in quantum yield for a difference of ~ 6.6 nm, while there is a ~ 16.1 nm difference between the optimal core-shell-shell structure and composition showcased in the paper, and the β phased version. Please provide further data to supplement and clear up this inconsistency or provide strong arguments as to why this comparison was made. The author should also provide a more consistent size comparison for the canonical structures as well.

4) The severe quenching due to the lack of protection from an inert shell is to be expected, but it would be interesting to see a comparison between the CY:10% Er@SYb structure and an unprotected codoped core (canonical structure/composition) and see the effect without removing the surface quenching aspect.

5) The finding that increasing Er³⁺ concentrations continues to increase the upconversion emissions is fascinating. Why did the author choose 10% as the optimal composition if the luminescent output is higher in other showcased compositions? Furthermore, a comparison of a completely doped (100% Er³⁺) system would also be interesting. What are the quantum yield results for these compositions? If indeed a 10% Er³⁺ doping ratio provides the highest quantum yield, then this needs to be demonstrated with quantum yield data. Please provide the complete data set showing quantum yield measurements for these showcased compositions and consider a wider range.

6) The temperature dependent photoluminescent observations are very interesting. It would be appreciated if the author can include measurements showing the photoluminescence at higher temperatures and comment on the effect of temperature on the peak ratios.

7) The finding that this core-shell-shell structure can be a general strategy for enhancing upconversion luminescence is fascinating! There should be quantum yield comparisons for the LiYF₄ structures for consistency, to showcase this interesting discovery.

8) To further emphasize previously mentioned points, there needs to be a more consistent approach to comparisons. Why is there no quantum yield data comparing the Tm³⁺ and Tb³⁺ doped systems to their canonical counterparts? This dataset would be crucial for showcasing these amazing discoveries.

9) The author is unclear in their description of the canonical Tm doped upconversion nanoparticle system. Are these structures α or β phased particles? The author should be specific for consistency.

10) Is there a functional utility for these core-shell-shell structured upconversion nanoparticles? It would be appreciated if the author can use a simple demonstration to showcase these great enhancements.

Reviewer #3 (Remarks to the Author):

In this work, the authors attempted separating sensitizers (Yb^{3+} ions) and activators (Er^{3+} ions, Tm^{3+} ions) in different layers (ie shells) to enhance multiphoton upconversion emission. Even though the authors have carried out good amount of work and decent control experiments, it is to my opinion that the novelty of this work is not strong enough for publication in Nature Communications. Some experiments are inconsistent and not comparable, please see my further remarks below.

1. Separation of sensitizers and activators in different layers has been widely reported and accepted in Nd^{3+} sensitized systems, which showed superior upconvsn performance and less heating up effect. (Zhong, Yeteng, et al. *Advanced materials* 26.18 (2014): 2831-2837, Shao, Wei, et al. *Journal of the American Chemical Society* 138.50 (2016): 16192-16195, etc.) This idea of separating sensitizers and activators is not novel enough.

2. The authors should demonstrate and prove cross-relaxation (CR) between Yb^{3+} and Er^{3+} ions since the authors claimed that the intrinsic cross-relaxation energy loss in UCNPs has not yet been properly addressed (line 57-58, page 3). The reason of concentration quenching is commonly attributed to that the increase of concentration of activators will induce more CR between activators, instead of CR between activator and sensitizer the authors claimed in abstract.

3. In comparison of quantum yields of different UCNPs, $\beta\text{-NaYF}_4:10\%\text{Er}@NaYbF_4@NaYF_4$ showed lower QY than that of $\alpha\text{-NaYF}_4:10\%\text{Er}@NaYbF_4@NaYF_4$, which is different from the former reported that β -UCNPs showed stronger UC emission than α -UCNPs. The authors should explain the reason for this. The sizes of UCNPs of α and β phases are different, which is not comparable and hence an appropriate control of equal size should be selected. .

4. As shown in Figure 1.e, the emission of $\text{C}_Y:10\%\text{Er}@S_{Yb}@S_Y$ and $\text{C}_Y:20\%\text{Yb}, 10\%\text{Er}@S_{Yb}@S_Y$ did not differ too much, which is not consistent with the authors' claim.

5. The authors tried to prove the enhancement of multiphoton upconversion emission, especially 407 nm. However, the significance of such enhancement is not fully emphasized, and one potential application should be demonstrated in this communication to show that after enhancement the nanostructured nanoparticles could still be appropriately applied to a particular field.

Response to reviewers' comments

Firstly, we would like to thank all reviewers for their precious comments, which helped us to improve the work dramatically.

The major changes to manuscript in this revision are briefly listed as follows:

1. The comparison of peak ratios was replaced with relative intensity of emission bands in Figure 1f. The pictures were taken with and without filters again for consistence and Figure 1g was updated.
2. NaYbF₄ shell thickness-dependent UCQY with power dependent UCQY was supplemented in Figure 2b. Figure 2d was updated with error bars added.
3. All possible backward energy transfer pathways are specified in updated Figure 3a. Figure 3c was updated with error bars supplemented. Downconversion luminescence of core-shell-shell nanoparticles was measured and supplemented in Figure 3d.
4. As a proof of concept experiment, we tested singlet oxygen (¹O₂) production activity of hematoporphyrin monomethyl ether (HMME, a commercial photosensitizer in photodynamic therapy) upon exciting the UCNPs to give an example for potential application. The results are shown in Figure 4. The original content in Figure 4 was partly moved to Supplementary Information.
5. Size deviation and UCQY of newly synthesized β-NaYF₄:10%Er@NaYbF₄@NaYF₄ were supplemented in Table 1. The reason for lower UCQY of β-NaYF₄:10%Er@NaYbF₄@NaYF₄ UCNPs was specified.
6. Two references including ref. 21 and 27 were added in the revised text.
7. Details of technical information about measurements at various temperatures and ¹O₂ production activity test were supplemented in Supplementary

Information. Supplementary Table 1 and 2 were added to summarize precursor dosage for synthesizing various UCNPs.

8. TEM-EDX of the core nanoparticles was recorded to prove the doping level of Er, and the result was supplemented in Supplementary Fig. 3. Yb lifetimes vs. rising Er concentration were added as Supplementary Fig. S15.
9. Quantum yields of various core-shell-shell nanoparticles were measured and the results were supplemented in Supplementary Table 4. A standard core-shell sample and $\alpha\text{-NaYF}_4\text{:10\%Er@NaYF}_4\text{@NaYbF}_4\text{@NaYF}_4$ nanoparticles were also synthesized (Supplementary Fig. 31 and 17) for comparison.
10. Fitting results of the decay curves at 407 nm (Supplementary Fig. 16a) for $\alpha\text{-NaYF}_4\text{:Er (2-100mol\%)@NaYbF}_4\text{@NaYF}_4$ core-shell-shell nanoparticles were supplemented in Supplementary Table 3 for better understanding the suppression of concentration quenching.
11. The UCL spectra for the $\alpha\text{-NaYF}_4\text{:10\%Er@NaYbF}_4\text{@NaYF}_4$ nanoparticles at higher temperatures were measured and supplemented in Supplementary Fig. 22.
12. Spectral sensitivity corrected UCL spectra for $\alpha\text{-NaYF}_4\text{:20\%Yb, 2\%Er(Tm)@NaYF}_4$ core-shell and $\text{NaYF}_4\text{:Er(Tm)@NaYbF}_4\text{@NaYF}_4$ core-shell-shell nanoparticles were added in Supplementary Information (Page 4-5, Supplementary Fig. 5 and 26e-f).

In the following pages, the reviewers' comments were replied point to point with the response highlighted in blue.

Response to referees' comments

Reviewer #1

The article by Ying Ma and colleagues entitled "Enhancing multiphoton upconversion through interfacial energy transfer in multilayered nanoparticles" presents interesting

observation of enhanced upconversion (UC), as soon as the sensitizer ions (Yb) are displaced from activators (Er, and some examples of Tm and Tb are provided as well). The article is relatively well written, in most cases the graphs are easily readable and clear. The novelty of this idea is deserving presentation, but there are however some shortcomings, which have to be addressed:

Response: We would like to thank the reviewer for positive evaluation and valuable comments that helped us to improve the paper greatly.

1. technical information about measurements in various temperatures is missing

Response: We have supplemented the technical information about measurements at various temperatures in Supplementary Information (Page 4). The samples for temperature-dependent PL measurements were prepared using a drop-casting method on a quartz glass substrate. We used an Oxford Optistat DryBL4 cryostat and a Microstat HiRes2 with a temperature controller (MercuryITC) to lower (3.8~300 K) and elevate temperature (300~500 K), respectively. During the test, the samples were held at a certain temperature for at least 10 min to assure equilibration.

2. measuring QY requires power dependent values and not single points. Moreover, I am also using FLS980 for QY measurements, and it is difficult (if not impossible) to get reliable values for UCQY on this equipment. Did the authors validate their results vs some standard sample and literature data e.g. comparing the obtained results from the publications of group of U.Resch-Genger.

Response: We have measured power dependent QY for C_Y : 10%Er@S_{Yb}@S_Y NCs ($d_{S_{Yb}} = 8.3 \pm 0.7$ nm, $d_{S_Y} = 2.3 \pm 0.1$ nm) and supplemented the results in Fig.2b (inset). We also synthesized β -NaYF₄: 18%Yb, 2%Er@NaYF₄ NCs ($d_{core} = 11.2 \pm 0.8$ nm, $d_{core-shell} = 19.9 \pm 1.1$ nm, Supplementary Fig. 31a-b) according to U. Resch-Genger's report (U. Resch-Genger, et al. *Angew. Chem. Int. Ed.* 2018, 57, 1–6) as a standard sample and measured its QY upon excitation by 980 nm at 4.5 W/cm² to validate our results. The detected QY is 2.11 ± 0.72 , comparable to U. Resch-Genger's value (~ 2 at 4~5 W/cm²). These results have been added in Supplementary Table 4. As the reviewer mentioned, it is indeed difficult for us to

acquire reliable UCQY values for those samples with UCQY smaller than 0.1% on FLS980. For those samples with UCQY higher than 0.1%, we always acquired a relatively stable value whenever we repeated the measurement.

3. it is not clear if the authors used the same concentration of NPs in all experiments to make reliable comparison. There is no technical information on this. How this was achieved?

Response: We used the similar concentration of NPs in all experiments except for quantum yield measurements. For making reliable comparison, all kinds of nanoparticles were prepared from 0.3 mmol core nanoparticles (details are shown in newly added Table S1 and S2) and dispersed in 3 mL of cyclohexane for measurement. It should be noted that the concentrations of core-shell-shell and core-shell-shell-shell nanoparticles might be smaller than that of core-shell nanoparticles since the inevitable loss would have occurred during shell growth, while nearly the same concentration would be obtained for various core-shell-shell NCs. For those UCNPs with the same size, the concentrations of lanthanide ions are nearly the same in the dispersion. Such details have also been supplemented in Supplementary Information (Page 3-4).

4. Discussing the impact core-shell-shell NPs on their spectral properties, more detailed work is necessary to demonstrate the actual composition of dopants is exactly as the authors intended to have. Showing just Fig.1c for single NPs is far too little. Moreover, the 10%Er is big enough to be detected with TEM-EDX - and this must be done to convince readers about reliability of results and materials.

Response: We have supplemented TEM-EDX results for α -NaYF₄: 10%Er NCs in Supplementary Fig. 3f (the inset). The detected Er content is 9.27±1.12%, similar to the nominal value.

5. There are many samples and tests, but in most cases, there is not enough information linked to the graphs, which precise which sample is used. Just for example - Fig.1 - which sample (in terms of shell thickness) was used here. Such information is missing also for some other graphs and must be corrected/supplemented with proper information.

Response: Sample information including thickness of NaYbF₄ shell and NaYF₄ shell has been supplemented in all figures and tables including those in Supplementary Information.

6. The spectra are not corrected for spectral sensitivity (which includes grating, PMT, optics etc.) - this is not a problem for relative changes, but is crucial if the authors claim the UC emission is X times larger than the green or red one.

Response: Taking Tm³⁺ doped nanoparticles as an example, versatile UCL spectra have been reported previously (Figure 1-2, below). Considering our work here mainly focuses on the enhancement of UC emission from higher-energy levels, we collected all spectra with the mode of “emission correction off”, as adopted by some authors (Figure 2). To avoid confusing the readers, we have provided an explanation on spectral sensitivity correction and supplemented corrected spectra for α -NaYF₄:20%Yb, 2%Er(Tm)@NaYF₄ core-shell ($d_{SY} = 10.5 \pm 0.8$ nm) and NaYF₄:Er(Tm)@NaYbF₄@NaYF₄ core-shell-shell ($d_{SYb} = 8.3 \pm 0.7$ nm, $d_{SY} = 2.3 \pm 0.1$ nm) nanoparticles in Supplementary Information (Page 4-5, Supplementary Fig. 5 and 26e-f).

Figure 1. Left: Comparison of upconversion spectra of the as-synthesised NaYF₄:Yb/Tm nanocrystals with different Tm³⁺ concentrations excited at 10 W/cm² (Fig S2a, Zhao, J. et al. *Nat. Nanotechnol.* 2013, **8**, 729-734). Right: Upconverting emission spectrum of β -NaGdF₄: 70 mol %Yb³⁺, 1 mol %Tm³⁺ @ β -NaGdF₄ core-shell UCNPs (60 μ g/mL) in cyclohexane at room temperature upon irradiation by 980 nm NIR laser (1 W/mm², Figure 3b, Wang, L. et al. *J. Am. Chem. Soc.* 2014, **136**, 4480-4483).

Figure 2. Emission intensity comparison of the $\text{NaYF}_4@ \text{NaYbF}_4:\text{Tm}(1\%)@ \text{NaYF}_4$ and $\text{NaYF}_4:\text{Yb/Tm}(40/1\%)@ \text{NaYF}_4$ nanoparticles. The spectra were recorded under excitation of a 980-nm CW diode laser at a power density of 20 W cm^{-2} (Supplementary Fig. 6c, Chen, X. et al. *Nat. Commun.* 2016, 7, 10304).

In this case, it is really not suitable for us to claim the UC emission of the green is X times larger than the red one. Thus, we have redrawn Fig. 1f, in which peak ratios (I_{407}/I_{540} and I_{651}/I_{540}) have been replaced by relative UCL intensity of violet, green and red emission bands. We have deleted the discussion about the peak ratios in the revised text.

7. *How the shell thickness was measured and what was the accuracy of this determination? What are the error bars at the graphs (e.g. Fig.2b and d) - look at my comment on spectral correction as well.*

Response: The shell thickness was estimated through measuring 100 particles in TEM images of UCNPs before and after shell growth (details are shown in Supplementary Table 1). The error bars have been supplemented in Fig.2b and d. The standard deviation has been supplemented in each TEM image for estimating the size of UCNPs. According to your suggestions, NaYbF_4 thickness dependent UCQYs and relative UCL intensity changes of the three emission bands are shown in Fig. 2b and d, respectively, in the revised manuscript.

8. *Actually, to make the results on Fig.2b meaningful, not (or not only) the intensity versus d_{SYb} should be presented, but the UCQY to account for rising Yb concentration.*

Response: We have replaced Fig.2b with NaYbF₄ thickness dependent UCQYs.

9. If only filter was exchanged in front of the camera for pictures 1g, why the positions of the laser beam within the sample is at different heights? Why there is more scattering in bottom pictures on Fig.1g? This is also important for UCQY measurements as was shown by Pilch et al. in YbHo UCNPs. What was the acquisition time for the pictures? Was it the same for all pictures ?

Response: The positions of the laser beam changed because the pictures were not taken at the same time. In fact, both samples scatter light because their particle sizes are larger than 30 nm. The stronger light the particles emit, the more scattering we can observe. We took the pictures again and supplemented the exposure time of 0.01 s in Supplementary Information (Page 4). Fig. 1g has also been updated in the revised manuscript. For UCQY measurements, all dispersions were diluted to minimize the effect of scattering.

10. Fig.3 - the energy transfer diagram should be confronted with recent papers from M. Bery and E. Chan for the UC and BET. It seems to me not all possible and probable BET mechanisms are shown (e.g. 4I11/2 Er  2F5/2 Yb). Again, which shell thicknesses are presented? It would be highly interesting to see not only UV/Vis emission but also 1550 nm emission from Er in this new configuration. Panel d does not really mean anything useful for current discussion.

Response: The energy transfer diagram has been redrawn, in which three possible BET pathways proposed from M. Berry (*J. Phys. Chem. C* 2017, 121, 16592-16606) and two possible BET pathways proposed from E. Chan (*Nano Lett.* 2016, 16, 7241-7247) are included (Totally four possible BET pathways are shown since one is proposed duplicatedly). These references have been supplemented in the revised text. The information about shell thickness ($d_{SYb} \geq 3.2$ nm, $d_{SY} \geq 2.0$ nm) has been supplemented in caption of Fig.3a.

According to the reviewer's suggestion, 1550 nm emission from Er in core-shell-shell nanoparticles was also recorded and shown in Fig. 3d and Supplementary Fig. 15 in the revised version. The results show that downshifting emission intensifies constantly with the increment of the sensitizers (d_{SYb}). On

average, the amount of Yb^{3+} ions increases by ~22-fold when d_{SYb} varies from 1.3 to 8.3 nm, with the near Infrared (NIR) emission intensity increasing by ~30-fold correspondingly. This phenomenon validates the fact that upconversion efficiency increases without sacrificing downconversion efficiency, further verifying the efficient energy transfer in this structure. Related discussion has been supplemented in the text (Line 6-12, page 9).

How come the low Yb->Er ET (red curve, inset for Fig.3b) makes the UC brighter than the higher Yb->Er ET (corresponding black curve). Yb lifetime should show Yb spontaneous lifetime modified by both outgoing (Yb->Er ET), and incoming (Er->Yb ET) energy? I understand the lack of Er->Yb BET, but how the energy migrated to Er ions that sit deeper in the NP volume? Is there no CR between Er ions at all? Very often, the authors of the manuscript describe what they see, but very often I miss deeper explanation, why it happens so, especially, such interesting results are presented.

Moreover, the lifetimes (Fig.3b) are counter-intuitive to me: why larger [Yb] concentration (related to larger thickness of d_{SYb}), makes the rise and decays longer? Energy migration between Yb ions has been observed to apparently increase the lifetime of Yb^{3+} , but this was observed in bulk materials (at much longer distances). Moreover, the decays are bi-exponential for me - why? To me, authors should also relate their results to the results of Pilch et al. on the impact of chemical architecture of YbHo on the Ho lifetimes.

Response: In Yb/Er codoped structure, the long Yb lifetime means the low $\text{Yb} \rightarrow \text{Er}$ ET and shorter Yb lifetime corresponds to the higher $\text{Yb} \rightarrow \text{Er}$ ET. In this spatially separated configuration, much longer time is needed for excitation energy to migrate to the core-shell interface and then transferred to Er^{3+} ions because the Yb-Er distance is much longer for Yb^{3+} ions in the NaYbF_4 shell far away from the core. In this sense, Yb lifetime here represents the speed of $\text{Yb} \rightarrow \text{Er}$ ET instead of the efficiency of $\text{Yb} \rightarrow \text{Er}$ ET because most of the absorbed energy will be consumed after energy migration through the $\text{Yb} \rightarrow \text{Er}$ ET process at the core-shell interface. Similar upconversion dynamics tailoring via energy migration has been reported previously (Zuo, J. et al. *Angew. Chem. Int. Ed.* 2018, 57, 3054-3058, reference 22 in the text).

We believe there is CR between Er ions, which benefits energy migration or

hopping to Er ions that sit deeper in the NP volume. The shorter lifetime of Er at high doping levels is the direct evidence of CR between Er ions. At relatively low concentration (less than 30%), the prolonged population of Er³⁺ emitting states (Supplementary Fig. 16) may benefit energy migration via CR and hinder CR induced nonradiative decay pathways, evidenced by the disappearance of the fast decay component at 407 nm (Supplementary Table 3). Related discussion has been added in the revised text (Page 9-10, 13).

For Yb³⁺ ions neighboring the outmost inert shell, the Yb-Er distance is approximately equal to the thickness of NaYbF₄ shell, whereas, the longest Yb-Er distance is estimate to be less than 1.6 nm in bulk materials (20%Yb, 2%Er, on average, there exist one Yb³⁺ ion in 2.5 unit cells). For most of the core-shell-shell UCNPs presented in this work, NaYbF₄ shell thickness is larger than 1.6 nm. Thus, the lifetime of Yb³⁺ ions, the rise and decay times of Er³⁺ ions are longer. Similar decay dynamics has been observed and theoretically simulated in NaYF₄: 20% Yb, 2% Er@NaYF₄: 20% Yb nanostructures in above mentioned paper (*Angew. Chem. Int. Ed.* 2018, 57, 3054-3058).

We agree with the reviewer that the decay curve in the structure with the thinnest NaYbF₄ shell (~1.3 nm) follows bi-exponential kinetics. We think that the long decaying component resembles those with thicker NaYF₄ shell, and the fast decay of ²H_{9/2} state may come from Er³⁺ ions neighboring sensitizing layer because the protection layer is not thick enough herein (less than 4.0 nm). We have added related discussion in the revised text (Line 7-12, Page 8).

11. I don't understand why the authors mismatch alpha and beta phases, NaYF₄ and LiYF₄ hosts - this makes the message more complex, confusing and more difficult to follow. Moreover, either Tb and Tm samples are studied with such care as Er samples (lifetimes, EY etc.), or all this is moved to SI or skipped for clarity.

Response: It is difficult for us to control the epitaxial growth of β-NaYbF₄ shell (TEM images in Supplementary Fig. 23 and 31), while the growth of α-NaYbF₄ shell is much easier to control. In order to acquire reliable results, we chose α-NaYF₄ based UCNPs in this work. To preclude the possibility that this phenomenon only occurs in cubic phase, we also compare the UCL of Er³⁺ in β-NaYF₄ and LiYF₄ hosts. According to your suggestion, original Fig. 4 is deleted and related data are moved to SI in the revised version.

Moreover, Tm is very susceptible to CR both between Tm-Tm pairs and Tm-Yb, why 8% Tm seems to be not susceptible to Tm-Tm CR? While Yb-Er or Yb-Tm system may indeed show BET from activator to sensitizer and in consequence decrease ETU intensity, in Yb-Tb samples this cooperative process seems to be more complex - although down-shifting effect is known, much more details need to be studied before this Yb-Tb ETU in new "split" system is ready for publication. Moreover, again, the spectral sensitivity correction must be done, otherwise it looks like Tm is emitting in UV only, while in real, 800 nm emission is much stronger than Vis-but current Fig.4c presents uncorrected data and makes the comparison more difficult.

Response: As shown in Supplementary Fig. 26, the UCL of Tm at ~800 nm is indeed susceptible to CR between Tm-Tm pairs and exhibits a complex variation trend. Thus, we integrated UCL intensity in the range of 300-600 nm for accuracy in the revised version. We have abbreviated the discussion on Tm and Tb doped UCNPs since the detailed mechanism needs to be further investigated (Page 11). As the spectral sensitivity correction is concerned, we did not correct the spectra in order to emphasize the enhancement in multiphoton upconversion, as previous report did (Chen, X. et al. *Nat. Commun.* 2016, 7, 10304). We have supplemented an explanation on spectral sensitivity correction and corrected spectra for α -NaYF₄:20%Yb, 2%Er(Tm)@NaYF₄ core-shell ($d_{SY} = 10.5 \pm 0.8$ nm) and NaYF₄:Er(Tm)@NaYbF₄@NaYF₄ core-shell-shell ($d_{SYb} = 8.3 \pm 0.7$ nm, $d_{SY} = 2.3 \pm 0.1$ nm) nanoparticles in Supplementary Information (Page 4-5, Supplementary Fig. 5 and 26e-f).

12. Why the lifetimes rise from 2-30% of Er (Fig.S14) for rising [Er] concentration and then fall down? What are the error bars? Why there are bi-exponential decays? What do the rise times tell?

Response: The error bars have been supplemented in revised Supplementary Information. As shown in Fig. S16 (original Fig. S14), the rise times for 10% and 30% Er doped samples are much longer than the others, especially for the emission at 407 nm. We hypothesize that faster population of Er³⁺ emitting states may accelerate E³⁺-Er³⁺ interactions that lead to nonradiative decay pathways. As shown in Table S3, a fast decay component appears in all samples except for

10% and 30% Er doped samples and dominates at heavy doping level. This dynamics needs to be further investigated to give an accurate explanation.

Concluding - the overall results are very interesting and intriguing, but describing what one can see on the graphs is far too little and in-depth mechanism should be proposed (generally lack of Er->Yb BET is proposed) and additional experiments should be proposed to either support or reject this hypothesis (e.g. power dependent UCQY, TEM-EDX to prove real structure, Yb lifetimes vs rising Er concentration, Er QY vs displacement between Yb and Er shell with yet another empty shell). I know this suggestions are time consuming, but after reading the manuscript I cannot tell I understand what is going on inside. I suppose modelling such phenomena with rate equations may not be trivial (to account for core-shell), but maybe could help to understand the physics.

Response: We have measured power dependent UCQY (inset in Fig. 2b), TEM-EDX for the core (inset in Supplementary Fig. 3), Yb lifetimes vs. rising Er concentration (Fig. S15a), Er QY vs displacement between Yb and Er shell with yet another empty shell (Fig. S17 and Table S4). We are still trying to model such phenomena with rate equations due to the complex structure and hope to better understand the physics in the near future.

Based on our experimental results, we hypothesize that separation location for the Yb³⁺ ions and Er³⁺ ions may bring two effects on Yb/Er doped UCNPs: (1) This spatial distribution may alleviate Yb³⁺-Er³⁺ ion cross-relaxation; (2) Separation location may benefit energy migration via cross-relaxation between Er³⁺ ions and hinder cross-relaxation induced nonradiative decay pathways through prolonging the population of Er³⁺ emitting states. Consequently, concentration quenching would be suppressed in certain degree. Related discussion has been supplemented in revised text (Second paragraph, page 13).

Reviewer #2

This manuscript reports a two-pronged approach to enhancing upconversion nanoparticle quantum yield, claiming the highest reported quantum yield for these materials at low power density excitations. This is truly a remarkable find, especially

if the findings provide such a generalizable strategy as is claimed in the manuscript; however, there are a few concerns regarding some of the background theory, the suggestions implied by the mechanism, some technical details, and some potentially missing data points that would fully round out the manuscript and meet the standards of a Nature Communications publication:

Response: Thank the reviewer's positive evaluation and the comments, which helped us to improve the paper.

1) The author stated that the rationale, at least partially, for this core-shell-shell structure is that it would alleviate the backward energy transfer pathway in these upconverting nanoparticles systems. Is this pathway significant enough to address directly? It would be appreciated if the authors can supply a stronger rationale and supporting literature to back up this point. Checking citations 11 and 20 as cited by the author, it is not clear that this pathway is so detrimental to the author's goals, and as a matter of fact, it is implied that this back transfer actually assists stronger red and blue emissions, as is shown later in the manuscript.

Response: The backward energy transfer is one major cause for nonradiative energy loss in addition to cross-relaxation between activator ions. We have added all possible backward energy pathways in Fig. 3a and a literature (ref. 21) in the revised text, in which the detrimental effect of backward energy transfer on population of Er³⁺ energy levels was discussed. Besides backward energy transfer pathway “Yb³⁺(²F_{7/2}) + Er³⁺(⁴G_{11/2}) → Yb³⁺(²F_{5/2}) + Er³⁺(⁴F_{9/2}) assisting red emission, the other three pathways mainly lead to nonradiative energy loss or downconversion emission. Even for the former pathway, backward energy transfer also increase the possibility of nonradiative energy loss if the energy could not be transferred to Er³⁺ for the second time.

2) The author mentions a short Yb³⁺-Er³⁺ distance at the interface, yet later in the manuscript they state that "the rise in the decay times for the Er³⁺ emission are obviously prolonged" due to the same structural feature. This seems inconsistent, and careful wording or explanation would be appreciated.

Response: In our activator doped core@sensitizer shell@inert shell structure, the

distance of Yb^{3+} - Er^{3+} at the interface between NaYF_4 :10%Er core and NaYbF_4 shell is shorter than that in the canonical structure because higher concentration of Er^{3+} ions lie in the core side and a large amount of Yb^{3+} ions exist in the shell side. For other regions far away from the interface, only Er^{3+} ions or Yb^{3+} ions exist and the distance of Yb^{3+} - Er^{3+} is much longer, so the excitation energy from the sensitizing shell far away from the interface can be only transferred to Er^{3+} ions at the interface after the energy is transported to the interface through energy migration (i.e. Yb-Yb cross relaxation).

3) The author states that the canonical structure for UCNPs is a codoped NaYF_4 system, which is true; however, my impression is that the canonical structures are usually β -hexagonal as opposed to α phase used in this paper. It would be therefore important to include the β phase canonical structure in your quantum yield measurements. Furthermore, checking your quantum yield comparisons table (Supplementary Table 1), it is interesting to note that there is a large size discrepancy between the β phased particles and the α phased particles. This is of particular concern considering that within the α phase particles, there is a ~1% difference in quantum yield for a difference of ~6.6 nm, while there is a ~16.1 nm difference between the optimal core-shell-shell structure and composition showcased in the paper, and the β phased version. Please provide further data to supplement and clear up this inconsistency or provide strong arguments as to why this comparison was made. The author should also provide a more consistent size comparison for the canonical structures as well.

Response: We have synthesized β - NaYF_4 :18%Yb,2%Er@ NaYF_4 UCNPs according to U. Resch-Genger's report (U. Resch-Genger, et al. *Angew. Chem. Int. Ed.* 2018, 57, 1–6, ref 7 in Supplementary Information) and measured their quantum yield for comparison (Supplementary Table 4). The detected quantum yield for this sample is comparable to the reported value. Both the reported value and our detected value are listed in Table S4. For our β phased particles with trilayers, the size difference is one reason leading to lower quantum yield than α phased particles. The other reason may be the anisotropic growth of β - NaYbF_4 shell, as demonstrated by TEM observation (Supplementary Fig. 23b). In fact, the length of these nanorods is larger than α phased particles, but their diameter is smaller than α phased particles. Thus, we modified the synthesis

method slightly and acquired nanoparticles with isotropic structure. Unfortunately, we found the growth of β -NaYbF₄ shell is still uncontrollable in this work, which was demonstrated by TEM observation after shell growth (Supplementary Fig. 31d). There are smaller nanoparticles with dark core, indicating partly unsuccessful epitaxial growth of NaYbF₄. Therefore, the detected quantum yield is ~4.99%, smaller than the corresponding α phased particles. We believe the lower quantum yield is an artifact induced by our synthesis. We have specified the reason for lower quantum yield of β phased particles in the revised manuscript (Table 1).

4) The severe quenching due to the lack of protection from an inert shell is to be expected, but it would be interesting to see a comparison between the CY:10% Er@SYb structure and an unprotected codoped core (canonical structure/composition) and see the effect without removing the surface quenching aspect.

Response: We supplemented comparison between the CY:10%Er@SYb structure and an unprotected codoped core in Supplementary Fig. 4. Obviously, much stronger emission can also be observed from the former than the latter. In fact, NaYbF₄ shell acts as not only a sensitizing layer but also an inert layer for Er³⁺ emission in this case since emission from Yb³⁺ is much less sensitive to surface quenching. In other word, there still exists a protection layer in CY:10%Er@SYb structure, different from unprotected codoped core.

5) The finding that increasing Er³⁺ concentrations continues to increase the upconversion emissions is fascinating. Why did the author choose 10% as the optimal composition if the luminescent output is higher in other showcased compositions? Furthermore, a comparison of a completely doped (100% Er³⁺) system would also be interesting. What are the quantum yield results for these compositions? If indeed a 10% Er³⁺ doping ratio provides the highest quantum yield, then this needs to be demonstrated with quantum yield data. Please provide the complete data set showing quantum yield measurements for these showcased compositions and consider a wider range.

Response: The quantum yields for core-shell-shell nanoparticles with Er³⁺ concentration increasing from 2 to 100% are supplemented in Table S4. The

α -NaYF₄:10%Er@NaYbF₄@NaYF₄ UCNPs exhibit the highest quantum yield. The corresponding UCL spectrum, downconversion luminescence spectrum and decay curves of α -NaErF₄@NaYbF₄@NaYF₄ have also been supplemented in Supplementary Information (Supplementary Fig. 14-16).

6) *The temperature dependent photoluminescent observations are very interesting. It would be appreciated if the author can include measurements showing the photoluminescence at higher temperatures and comment on the effect of temperature on the peak ratios.*

Response: The photoluminescence of the α -NaYF₄:10%Er@NaYbF₄@NaYF₄ nanoparticles at higher temperatures has been supplemented in Supplementary Fig. 22. The temperature dependent spectra demonstrate obvious thermal quenching effect. Small abnormal intensity increase at temperatures higher than 488 K may be due to surface defects minimization. Red emission becomes dominant at higher temperatures due to quickly quenching of the emission band around 407 nm.

7) *The finding that this core-shell-shell structure can be a general strategy for enhancing upconversion luminescence is fascinating! There should be quantum yield comparisons for the LiYF₄ structures for consistency, to showcase this interesting discovery.*

Response: Quantum yield comparisons for the LiYF₄ structures are supplemented in Table S4. The LiYF₄:10%Er@LiYbF₄@LiYF₄ nanoparticles present a higher quantum yield of ~0.90%, while the LiYF₄:20%Yb, 2%Er@LiYF₄ nanoparticles exhibit a lower quantum yield of ~0.23%. These results further validate the superiority of this core-shell-shell structure.

8) *To further emphasize previously mentioned points, there needs to be a more consistent approach to comparisons. Why is there no quantum yield data comparing the Tm³⁺ and Tb³⁺ doped systems to their canonical counterparts? This dataset would be crucial for showcasing these amazing discoveries.*

Response: We also measured the quantum yields for Tm³⁺ and Tb³⁺ doped

nanoparticles and only acquired the value for α -NaYF₄:8%Tm@NaYbF₄@NaYF₄ (Table S4, 0.48±0.04%). The quantum yields of the other three are smaller than 0.1%, which is lower than the detection limit of our spectroscopy.

9) The author is unclear in their description of the canonical Tm doped upconversion nanoparticle system. Are these structures α or β phased particles? The author should be specific for consistency.

Response: For Tm doped UCNPs, the host is α phase, too. We have specified all the samples (the canonical α -NaYF₄: Yb/Tm@NaYF₄ nanoparticles) in the text, figure captions and tables in the revised version.

10) Is there a functional utility for these core-shell-shell structured upconversion nanoparticles? It would be appreciated if the author can use a simple demonstration to showcase these great enhancements.

Response: The intensified emission from high-energy levels may offer more opportunities for deep-tissue biophotonics, such as photodynamic therapy and optogenetics. As a proof of concept experiment, we tested singlet oxygen (¹O₂) production activity of hematoporphyrin monomethyl ether (HMME, a commercial photosensitizer in photodynamic therapy) upon exciting the UCNPs to validate our conjecture. The results are supplemented in Fig. 4 and Fig. S32-33. The experimental results show that ¹O₂ production activity of C_Y:2% Er@S_{Yb}@S_Y UCNPs reaches 3.8 times that of C_Y:20% Yb, 2% Er@S_Y UCNPs. Considering the neglected enhancement in green and red emission for C_Y:2% Er@S_{Yb}@S_Y UCNPs as compared with C_Y:20% Yb, 2% Er@S_Y UCNPs, the activity promotion should be attributed to enhanced violet emission. Related discussion is also supplemented in the text (last paragraph, page 13-14).

Reviewer #3

In this work, the authors attempted separating sensitizers (Yb³⁺ ions) and activators (Er³⁺ ions, Tm³⁺ ions) in different layers (ie shells) to enhance multiphoton upconversion emission. Even though the authors have carried out good amount of

work and decent control experiments, it is to my opinion that the novelty of this work is not strong enough for publication in Nature Communications. Some experiments are inconsistent and not comparable, please see my further remarks below.

Response: Thank for the reviewer's evaluation and comments helping us to improve the manuscript.

1. Separation of sensitizers and activators in different layers has been widely reported and accepted in Nd³⁺ sensitized systems, which showed superior upconversion performance and less heating up effect. (Zhong, Yeteng, et al. Advanced materials 26.18 (2014): 2831-2837, Shao, Wei, et al. Journal of the American Chemical Society 138.50 (2016): 16192-16195, etc.) This idea of separating sensitizers and activators is not novel enough.

Response: It is well known that ⁴I_J manifolds of Nd³⁺ ion make it very efficient for excitation energy to backward transfer to Nd³⁺ ions from different activators, so separation of Nd³⁺ ions and activators is necessary and widely accepted for construction of Nd³⁺ sensitized UCNPs. Different from Nd³⁺ ion, Yb³⁺ ion has two energy levels (²F_{7/2} and ²F_{5/2}) and backward energy transfer to Yb³⁺ ions from the activators has long been neglected. In order to assure efficient Yb³⁺→activator energy transfer, Yb³⁺ ions and activators are usually codoped in the same layer for keeping a short Yb-activator distance. Even for the two structures adopted in above mentioned two papers, NaYF₄:Yb,Er@NaYF₄:Yb@NaNdF₄:Yb, and NaYF₄:Yb³⁺/X³⁺@NaYbF₄@NaYF₄:Nd³⁺ (X = Er, Ho, Tm, or Pr), Yb³⁺ ions are still codoped with activator ions in the core and with Nd³⁺ ions in the shell for efficient Yb³⁺→activator and Nd³⁺→Yb³⁺ energy transfer, respectively, totally different from the concept of our structure. Our results demonstrate that the backward energy transfer from the activators to Yb³⁺ ions also contributes a lot to excitation energy loss in Yb³⁺ sensitized UCNPs. We provide here an efficient strategy to alleviate such a detrimental effect and improve brightness of UCNPs.

2. The authors should demonstrate and prove cross-relaxation (CR) between Yb³⁺ and Er³⁺ ions since the authors claimed that the intrinsic cross-relaxation energy loss in UCNPs has not yet been properly addressed (line 57-58, page 3). The reason of concentration quenching is commonly attributed to that the increase of concentration

of activators will induce more CR between activators, instead of CR between activator and sensitizer the authors claimed in abstract.

Response: Generally, the concentration quenching induced by elevating the concentration of the activators is widely accepted, the contribution of CR between activator and sensitizer (mainly for Yb^{3+}) to concentration quenching has long been neglected. First, we demonstrate here that all of the multilayered structures with Yb^{3+} and Er^{3+} codoped into the core present a much lower UC luminescent intensity than our $\text{C}_Y:10\% \text{Er}@_{\text{S}_{\text{Yb}}}\text{S}_Y$ structure despite their particle sizes, sensitizer layers and inert layers all being nearly the same (Fig. 1). In particular, $\text{C}_Y:10\%\text{Er}@_{\text{S}_{\text{Yb}}}\text{S}_Y$ sample presents much more stronger emission than $\text{C}_Y:20\%\text{Yb}, 10\%\text{Er}@_{\text{S}_{\text{Yb}}}\text{S}_Y$ sample. There are more sensitizer ions in the latter, but UCL intensity decreases, especially for that at ~ 407 nm. It is the only difference between the two structures that Yb^{3+} is codoped in the core for $\text{C}_Y:20\%\text{Yb}, 10\%\text{Er}@_{\text{S}_{\text{Yb}}}\text{S}_Y$. Secondly, comparison UCL of the three 2% Er-doped UCNPs in Figure 3c provides clear evidence for CR between Yb^{3+} and Er^{3+} . The three-photo upconversion is enhanced by 4-fold in $\text{C}_Y:2\%\text{Er}@_{\text{S}_{\text{Yb}}}\text{S}_Y$ UCNPs as compared with $\text{C}_Y:20\%\text{Yb}, 2\%\text{Er}@_{\text{S}_{\text{Yb}}}\text{S}_Y$. Considering more excitation energy is needed for three-photo upconversion, energy loss due to codoping of Yb^{3+} and Er^{3+} in the core is obvious. Thirdly, Supplementary Fig. 14 and 16 show that the UCL intensity for this trilayered structure decreases when Er concentration reaches 70%, and the life times of Er emitting states increases with Er concentration increasing from 2% to 30%. In previously reported work (*Nat. Nanotechnol.* 2014, 9, 300-305), codoping of Yb^{3+} and Er^{3+} leads to a reduced lifetime with increasing Er^{3+} content, indicating obvious concentration quenching. We believe that these results prove cross-relaxation (CR) between Yb^{3+} and Er^{3+} ions.

3. *In comparison of quantum yields of different UCNPs, $\beta\text{-NaYF}_4:10\%\text{Er}@_{\text{NaYbF}_4}\text{NaYF}_4$ showed lower QY than that of $\alpha\text{-NaYF}_4:10\%\text{Er}@_{\text{NaYbF}_4}\text{NaYF}_4$, which is different from the former reported that $\beta\text{-UCNPs}$ showed stronger UC emission than $\alpha\text{-UCNPs}$. The authors should explain the reason for this. The sizes of UCNPs of α and β phases are different, which is not comparable and hence an appropriate control of equal size should be selected.*

Response: As shown in Supplementary Fig. 23, the diameter of our β -NaYF₄:10%Er@NaYbF₄@NaYF₄ nanorods ($26.1\pm 4.3 \times 37.2\pm 3.9$) is really smaller than α -NaYF₄:10%Er@NaYbF₄@NaYF₄ particles (32.2 ± 2.2), but their length is larger than the latter. Thus, we think the anisotropic growth of β -NaYbF₄ shell may contribute more to their lower quantum yield. We tried to synthesize larger β -NaYF₄:10%Er@NaYbF₄@NaYF₄ particles, but we found the growth of β -NaYbF₄ shell is still uncontrollable. There is a large deviation in the sizes of β -particles and independent growth of NaYbF₄ also appears (Supplementary Fig. 31d). Consequently the detected quantum yield of newly synthesized nanoparticles is also lower than α phased particles. We believe the lower quantum yield is an artifact induced by our synthesis. We have specified the reason for lower quantum yield of β phased particles in the revised manuscript (Table 1).

4. As shown in Figure 1.e, the emission of C_Y:10%Er@S_{Yb}@S_Y and C_Y:20%Yb, 10%Er@S_{Yb}@S_Y did not differ too much, which is not consistent with the authors' claim.

Figure 3. Comparison of normalized UCL spectra for C_Y:10%Er@S_{Yb}@S_Y and C_Y:20%Yb, 10%Er@S_{Yb}@S_Y UCNPs.

Response: We normalized the two spectra by the red emission band for better comparison. As shown in Figure 3, the intensity of the emission band at ~407 nm

for $C_Y:20\%Yb, 10\%Er@S_{Yb}@S_Y$ decrease almost half as compared with $C_Y:10\%Er@S_{Yb}@S_Y$, while their green and red emission intensities are nearly the same. We think this difference provides a direct evidence for the detrimental effects of Yb-Er cross-relaxation on UC emission.

5. The authors tried to prove the enhancement of multiphoton upconversion emission, especially 407 nm. However, the significance of such enhancement is not fully emphasized, and one potential application should be demonstrated in this communication to show that after enhancement the nanostructured nanoparticles could still be appropriately applied to a particular field.

Response: Considering the intensified emission from high-energy levels may offer more opportunities for deep-tissue biophotonics, such as photodynamic therapy and optogenetics, we tested singlet oxygen (1O_2) production activity of hematoporphyrin monomethyl ether (HMME, a commercial photosensitizer) upon exciting the UCNPs to validate our conjecture (last paragraph, page 13, Fig. 4 in the revised manuscript, page 5 and Supplementary Fig. 32-33 in SI). The augmentation of singlet oxygen sensor green reagent (SOSG) fluorescence in aqueous dispersion of $C_Y:20\% Yb, 2\% Er@S_Y$ UCNPs is about 12.3% after 980 nm laser irradiation for about 30 min (Figure 4b and Supplementary Fig. 33). In contrast, the SOSG fluorescence augmentation is about 46.4% and 61.5%, respectively, in aqueous dispersion of $C_Y:2\% Er@S_{Yb}@S_Y$ and $C_Y:10\% Er@S_{Yb}@S_Y$ UCNPs. While the fluorescence of the control sample decreases about 2.8%. As compared with $C_Y:20\% Yb, 2\% Er@S_Y$ UCNPs, neglected enhancement in green and red emission for $C_Y:2\% Er@S_{Yb}@S_Y$ UCNPs was observed (Figure 3c), thus the improved 1O_2 production activity for the latter two samples should be attributed to their strong violet emission.

Reviewers' Comments:

Reviewer #1:

Remarks to the Author:

I appreciate the response from the authors, who improved the manuscript significantly. There are however still some points to clarify and add, before the MS is ready for acceptance.

1. Why Fig.2b stops at $d_{SYb}=10\text{nm}$ as the curve continuously rises up ? The same question is valid for the inset in Fig.2b, Fig.3d, 3b? Such unprecedented high values of QY deserves versatile characterisation even at higher thickness of Yb intermediate shell as well as excitation power in this case. If this is going to "change the game", the authors cannot stop "in the middle of the trip".
2. I have requested to study TEM-EDX maps - i.e. the composition maps of the nano-crystals (similar to, but more detailed to the one in Fig.1c) - this is critically important for understanding the observed behaviour and relate it to the structure of the core-shell-shell composition. Why? For example, we have synthesized beta-NaYF₄ Yb@Er core shell materials for a project we have, and they were much weaker than YbEr or Yb@YbEr, which is in opposite to the claims made in this manuscript. Therefore, having 20%Yb and 10%Er of such high concentration makes them eligible for NC composition mapping with TEM-EDX (i.e. the concentrations are high enough) to check if the Yb and Er ions co-localize in some intermediate space between core and the first shell. Without this result, the data presented by the authors are just general observations and hypotheses only. Pure single Er doped up-conversion is of course possible, but is significantly enhanced with Yb sensitizer, which is intuitively correct when you know how APTES work. However, the claims made by the authors, that separating Er from Yb enhance the up-conversion are somehow against intuition and definitely require deeper understanding and more facts.
3. What is the mechanism behind the PDT in this MS ? Usually, this is FRET between Er donors and the PS acceptor, but here Er is inside the core, too far away (a few times Forster distance) from the PS acceptor at the surface. Therefore, this is most probably reabsorption of light, but this hypothesis would require validation (e.g. lifetimes vs concentration of PS for example). I understand the reason to put PDT experiment in this manuscript, but this, rather trivial demonstration, does not explain anything else than the fact, the new NPs are brighter than conventional core only NPs.
4. Actually I am astonished that it is more easy to control the shape and size in alpha phase NaYF₄ than in beta nano-crystals.

Reviewer #2:

Evaluation:

- Data is technically sound: Satisfactory data accomplished with the addition of significant key data points in the SI as well as the amendment of the text to provide more relevant figure sets.
- Paper provides strong evidence: The manuscript contains strong evidence and rationale.
- Results are novel: Results are sufficiently novel
- The manuscript is important to scientists in the specific field: Manuscript is sufficiently important to advancing the field.
- Recommendation: The novelty of the results and findings are sufficient and relevant to advancing the field of upconversion nanoparticles for consideration. Considerable for Nature Communications with minor revisions addressing the concerns laid out below.

The authors have given a strong effort at addressing the concerns of the reviewers. More specifically, they have added a small utility demonstration as well as significantly expanding certain parts of the SI to address the reviewer's concerns; however, while many changes are made, not all of them are positive, and there remains room for improvement.

- 1) It is not necessarily a bad/incorrect for the B phase to have a lower quantum yield in the case of your structure, but it is interesting to note and it would be interesting if there might be some other way to think about this enhancement. The clarification and addition of data are appreciated.
- 2) It might be an interesting addition to add a photograph of the main canonical compositions compared with your uniquely structured nanoparticle to showcase improvements.
- 3) The additional quantum yield data and the utility demonstration are greatly appreciated additions; however, the restructuring of the figures leaves the last one a bit lacking.
- 4) The proof of concept is a welcome addition to the manuscript, but it would probably be better served if it was expanded or if there was another figure more comprehensively capturing the scope of your work. For example, a more tailored version of the original figure 4 where figure 4 a is replaced by emission color photographs and certain key data points or perhaps a schematic is added to emphasize the results. Either figure 4 should be expanded or another figure should be added. As it stands, the figures as a whole are a bit lacking compared to other recent publications in Nature comm. This includes publications on similar upconversion nanoparticle-based topics.
- 5) While the authors mainly compare their structured UCNPs to a canonical standard, there have been many others doing similar work in recent times. For example, [Nature Communications 9, 3082 (2018)] and [Nature Photonics 12, 5488-553 (2018)] both demonstrate interesting core-shell strategies with increased Er concentrations for low power density luminescent enhancements while [Advanced Materials 31, 1806991 (2019)] also demonstrate a core-shell structure designed to minimize back energy transfer to provide emissions from low power density excitations. Can you comment on these recent advances and clarify the novelty of your structure? The authors should update and amend the introduction to reflect the current status of this field and the impact of their contribution and findings.

- 6) Some of the authors' names are not properly written or spelled. Please proofread the text for these minor errors.

Reviewer #3:

Remarks to the Author:

The authors have adequately addressed the comments and therefore i recommend acceptance of the manuscript.

Response to reviewers' comments

We thank the reviewers for their helpful comments. We supplemented experimental data and modified the manuscript according to the comments and suggestions.

The major changes to manuscript in this revision are briefly listed as follows:

1. Figure 1 showing the main topic of our manuscript was supplemented.
2. Figure 2 (original Figure 1) was redrawn with EDX mapping results supplemented.
3. The quantum yields for trilayered structure with thicker NaYbF₄ layers (d_{SYb}) as well as those for trilayered structure with $d_{\text{SYb}} = 8.3$ nm excited at higher laser powers were supplemented in Figure 3 (original Figure 2).
4. Luminescence decay curves of UC emission and NIR downshifting luminescence spectra for C_Y:10% Er@S_{Yb}@S_Y nanoparticles with larger d_{SYb} were supplemented in Figures 4b, 4d (original Figures 3b, 3d) and Supplementary Fig. 12.
5. Schematic diagram for loading HMME and producing ¹O₂ on the UCNPs was supplemented in Figure 5 (original Figure 4).
6. Hexagonal (β -phase) C_Y:10% Er@S_{Yb}@S_Y nanoparticles ($d_{\text{SYb}} \approx 8.3$ nm, $d_{\text{SY}} \approx 2.0$ nm) were re-synthesized and their quantum yield was supplemented in Table 1 and Supplementary Table 4.
7. A comparison of α -NaYF₄:10%Er@NaYbF₄@NaYF₄ nanoparticles and the recently reported new nanostructures was supplemented in Supplementary Fig. 32.
8. Luminescence decay curves for Tween 20 modified NaYF₄:Er@NaYbF₄@NaYF₄ nanoparticles ($d_{\text{SYb}} = 8.3 \pm 0.7$ nm, $d_{\text{SY}} = 2.3 \pm 0.1$ nm) before and after loading the photosensitizer were supplemented in Supplementary Fig. 35.

In the following pages, the reviewers' comments were replied point to point with the response highlighted in blue.

Reviewer #1 (Remarks to the Author):

I appreciate the response from the authors, who improved the manuscript significantly. There are however still some points to clarify and add, before the MS is ready for acceptance.

Response: We thank the reviewer for positive evaluation and valuable comments that helped us to improve the paper greatly.

1. Why Fig.2b stops at $d_{SYb}=10nm$ as the curve continuously rises up? The same question is valid for the inset in Fig.2b, Fig.3d, 3b? Such unprecedented high values of QY deserves versatile characterisation even at higher thickness of Yb intermediate shell as well as excitation power in this case. If this is going to "change the game", the authors cannot stop "in the middle of the trip".

Response: We have supplemented the experimental data for core-shell-shell UCNPs with $d_{SYb} = 10.2$ and 13.1 nm in Fig.3a, Fig.3b, Fig.4b, Fig.4d (original Fig.2a, Fig.2b, Fig.3b, Fig.3d) and Supplementary Fig.12. It is found that UCL intensity and quantum yield of the UCNPs decreases when the NaYbF₄ shell is further thickened (Fig.3a and 3b). Similarly, NIR downshifting luminescence for these UCNPs with thicker NaYbF₄ shells decreases in comparison with the UCNPs with $d_{SYb} = 8.3$ nm (Fig.4d). These phenomena indicate that efficient energy migration and transfer is limited within certain distance (~ 8 nm herein) for C_Y:10% Er@S_{Yb}@S_Y trilayered structure. Moreover, the lifetimes of the UC emission bands for these UCNPs with thicker NaYbF₄ shell (larger than 10.2 nm) are shorter than those for the UCNPs with $d_{SYb} = 8.3$ nm (Fig.4b and Supplementary Fig.12), suggesting that luminescent quenching may also play a key role in PL decay for these UCNPs. In contrast to the obvious fade of the green and red emission, the violet emission gradually reaches saturation when the NaYbF₄ shell is larger than 10.2 nm. This finding implies that the population of the ⁴F_{9/2} level through a triphotonic transition may be hindered when a much thicker NaYbF₄ shell is coated. Quantum yields detected at higher excitation power for trilayered structure with $d_{SYb} = 8.3$ nm have also been supplemented in the inset. The quantum yield reaches $\sim 14\%$ when the excitation power is larger than 13.6 W/cm². Related discussion has been supplemented on page 6 (line 9, 12-15), page 7 (line 21-22), page 8 (line 1-3, 15-16, 19-20), page 10 (line 3-4). Supplementary Fig.7 was also updated.

2. I have requested to study TEM-EDX maps - i.e. the composition maps of the nano-crystals (similat to, but more detailed to the one in Fig.1c) - this is critically

important for understanding the observed behaviour and relate it to the structure of the core-shell-shell composition. Why? For example, we have synthesized beta-NaYF₄ Yb@Er core shell materials for a project we have, and they were much weaker than YbEr or Yb@YbEr, which is in opposite to the claims made in this manuscript. Therefore, having 20%Yb and 10%Er of such high concentration makes them eligible for NC composition mapping with TEM-EDX (i.e. the concentrations are high enough) to check if the Yb and Er ions co-localize in some intermediate space between core and the first shell. Without this result, the data presented by the authors are just general observations and hypotheses only. Pure single Er doped up-conversion is of course possible, but is significantly enhanced with Yb sensitizer, which is intuitively correct when you know how APTES work. However, the claims made by the authors, that separating Er from Yb enhance the up-conversion are somehow against intuition and definitely require deeper understanding and more facts.

Response: We have supplemented TEM-EDX elemental mapping results in Figure 2d. The distribution of Er³⁺, Yb³⁺ and Y³⁺ ions confirms the presence of Er doped core, the NaYbF₄ intermediate shell and NaYF₄ outmost shell. Unfortunately, it is difficult to discern the intermediate space between core and the first shell because both strong Yb signal and weak Er signal result in poorer contrast. However, intermixing of Yb³⁺ and Y³⁺ ions at the interface between NaYbF₄ shell and the outmost NaYF₄ shell can be clearly discerned. In fact, intermixing of core and shell materials (or cations from different shells) has been found previously [Resch-Genger et al., J. Am. Chem. Soc. 140, 4922–4928 (2018); Chu et al., Nat. Photon. 12, 548-553 (2018)]. Supposing cations intermixing does not occur at the interface between core and the first shell given here, the composition at core/NaYbF₄ shell interface is approximately NaYF₄: 50%Yb, 5%Er if one NaYF₄:10%Er layer and one NaYbF₄ layer are considered. Such a doping concentration is higher than that for canonical codoped structure (20%Yb, 2%Er), making it possible for the former (core/NaYbF₄ shell interface) to possess shorter Yb-Er distance than that for the latter. As a result, more efficient APTES happens at the interface between core and NaYbF₄ shell.

According to our experimental results, C_Y:10% Er@S_{Yb} core-shell UCNPs emit much weaker UCL than C_Y:20%Yb, 2%Er@S_Y core-shell and C_Y:10% Er@S_{Yb}@S_Y core-shell-shell UCNPs (Figure 2e and Figure 3c). This finding indicates surface quenching significantly affects UCL of these UCNPs since their particle sizes are similar (27.7±2.4 and 32.0±2.5 nm for the former and the latter, respectively). In other words, highly efficient energy transfer upconversion can only be realized when the surface quenching is suppressed and negligible. Similarly, more severe surface quenching for beta-NaYF₄:Yb@Er core-shell material than that for YbEr or Yb@YbEr structure may lead to its much weaker UCL than the latter two structures. On the other hand, larger Yb-Er distance (the doping concentration in separated core-shell structure is the same as that in codoped one) for the former than the latter may also lead to its weak UCL.

3. What is the mechanism behind the PDT in this MS? Usually, this is FRET between Er donors and the PS acceptor, but here Er is inside the core, too far away (a few times Förster distance) from the PS acceptor at the surface. Therefore, this is most probably reabsorption of light, but this hypothesis would require validation (e.g. lifetimes vs concentration of PS for example). I understand the reason to put PDT experiment in this manuscript, but this, rather trivial demonstration, does not explain anything else than the fact, the new NPs are brighter than conventional core only NPs.

Response: We have supplemented luminescence decay curves for Tween 20 modified NaYF₄:Er@NaYbF₄@NaYF₄ nanoparticles ($d_{\text{SYb}} = 8.3 \pm 0.7$ nm, $d_{\text{SY}} = 2.3 \pm 0.1$ nm) before and after loading the photosensitizer, HMME (Supplementary Fig.35). The invariable lifetimes exclude Förster Resonance Energy Transfer (FRET) mechanism. PDT experiment herein demonstrates the trilayered nanoparticles do emit much stronger light at shorter wavelength and efficiently initiate photoreactions, which are necessary for their potential application in deep-tissue biophotonics such as optogenetics. Related discussion has been supplemented on page 15 (line 5-10).

4. Actually I am astonished that it is more easy to control the shape and size in alpha phase NaYF₄ than in beta nano-crystals.

Response: In our previous work, we found relatively smaller beta-NaYF₄ nanoparticles could be synthesized using lanthanide trifluoroacetates as the precursors, but the sizes of these nanoparticles vary from one batch to another. We had to repeat the experiments several or tens of times to acquire the same sized nanoparticles. Whereas, the sizes of small alpha-NaYF₄ nanoparticles synthesized using the same precursors are nearly invariable for different batches. We believe thermal decomposition of sodium trifluoroacetate (NaTFA) before beta-core formation varies from one batch to another because excess NaTFA is necessary for minimizing nanoparticle size of beta phase (NaTFA decomposes at 250 °C and beta-core forms at a temperature higher than 300 °C). In contrast, there is no excess NaTFA and the temperature for formation of alpha-core is much lower than that for beta-core, so thermal decomposition of NaTFA can be neglected and the synthetic condition is invariable for different batches. According to these experiences, we chose alpha-phase to investigate the structure-dependent energy transfer upconversion of Yb/Er doped NaYF₄ UCNPs in order to get reliable results. To make sure that the conclusion is also suitable for beta-phase NaYF₄, the widely used host lattice, we also synthesized beta-phase using the same method. Unfortunately, the growth of beta-NaYbF₄ layer seems to be more difficult to control than NaYF₄ counterpart. To clarify which one is more favorable for photon upconversion for this trilayered structure: alpha-phase or beta-phase, we tried to re-synthesize beta-phase trilayered nanostructures following a procedure reported by Steven Chu et al.

[Nature Photonics 12, 5488-553 (2018)] with a slight modification and measured their quantum yield. The detected quantum yield for these better controlled beta-phase UCNPs is higher than their alpha-phase counterparts (6.82 ± 0.50 versus 5.42 ± 0.43 , Table 1), indicating beta-phase is still the better host. Accordingly, Supplementary Fig. 31 has been updated.

Reviewer #2

The authors have given a strong effort at addressing the concerns of the reviewers. More specifically, they have added a small utility demonstration as well as significantly expanding certain parts of the SI to address the reviewer's concerns; however, while many changes are made, not all of them are positive, and there remains room for improvement.

Response: We are grateful for the reviewer's helpful comments about improving the manuscript and addressed the comments accordingly below.

1) It is not necessarily a bad/incorrect for the B phase to have a lower quantum yield in the case of your structure, but it is interesting to note and it would be interesting if there might be some other way to think about this enhancement. The clarification and addition of data are appreciated.

Response: We have re-synthesized beta-phase by a better controllable method to clarify this phenomenon. As we expected, the quantum yield for these newly synthesized beta-phase UCNPs is indeed higher than their alpha-phase counterparts (6.82 ± 0.50 versus 5.42 ± 0.43 , Table 1). Table 1, Supplementary Fig.31 and Supplementary Table 4 have been updated.

2) It might be an interesting addition to add a photograph of the main canonical compositions compared with your uniquely structured nanoparticle to showcase improvements.

Response: We have added Figure 1 in which the compositions and luminescence properties of the canonical nanoparticles and our trilayered nanoparticles are compared. The schematic diagram in original Figure 1 (Figure 2 in the revised version) was replaced by TEM-EDX mapping results.

3) The additional quantum yield data and the utility demonstration are greatly appreciated additions; however, the restructuring of the figures leaves the last one a bit lacking.

Response: We have redrawn the last figure (Figure 5 in the revised version) and supplemented schematic diagram for loading HMME and producing $^1\text{O}_2$ on the UCNPs.

4) *The proof of concept is a welcome addition to the manuscript, but it would probably be better served if it was expanded or if there was another figure more comprehensively capturing the scope of your work. For example, a more tailored version of the original figure 4 where figure 4a is replaced by emission color photographs and certain key data points or perhaps a schematic is added to emphasize the results. Either figure 4 should be expanded or another figure should be added. As it stands, the figures as a whole are a bit lacking compared to other recent publications in Nature comm. This includes publications on similar upconversion nanoparticle-based topics.*

Response: We have redrawn all figures to improve the quality and highlight the results. In particular, a schematic comparison of the canonical UCNPs with newly structured UCNPs has been added to emphasize the topic or the main scope of our work. Since only green emission varies slightly during laser irradiation, no significant variation in emission color photographs could be observed. A schematic diagram for loading HMME and producing $^1\text{O}_2$ on the UCNPs has been supplemented in figure 4 (Figure 5 in the revised version) providing experimental details and mechanism. There are totally five figures in the revised manuscript.

5) *While the authors mainly compare their structured UCNPs to a canonical standard, there have been many others doing similar work in recent times. For example, [Nature Communications 9, 3082 (2018)] and [Nature Photonics 12, 5488-553 (2018)] both demonstrate interesting core-shell strategies with increased Er concentrations for low power density luminescent enhancements while [Advanced Materials 31, 1806991 (2019)] also demonstrate a core-shell structure designed to minimize back energy transfer to provide emissions from low power density excitations. Can you comment on these recent advances and clarify the novelty of your structure? The authors should update and amend the introduction to reflect the current status of this field and the impact of their contribution and findings.*

Response: We have synthesized the structures appeared in above mentioned papers and compared the PL spectra of these structures. The results are attached below and supplemented in Supplementary Fig. 32. The so called alloyed structure $\text{NaYbF}_4: 20\% \text{Er} @ \text{NaYF}_4$ [$\text{C}_Y @ \text{S}_{Yb}: 2\% \text{Er} @ \text{S}_Y$, Nature Communications 9, 3082 (2018)] emits much weaker UCL than our trilayered structure, especially for violet and red emission. For $\text{NaYF}_4: 20\% \text{Yb} @ \text{NaYF}_4: x\% \text{Er} @ \text{NaYF}_4: 20\% \text{Yb}$ structure [Advanced Materials 31, 1806991 (2019)], obvious quenching occurs when the doping content of Er is higher than 2% (Figure S4 in the published paper). Even though we increase Yb content to 100% ($\text{C}_{Yb} @ \text{S}_Y: 10\% \text{Er} @ \text{S}_Y$), this structure still emits weak UCL and no increased multiphoton upconversion can be observed. As $\text{NaYF}_4: 20\% \text{Yb} @ \text{NaYF}_4: 2\% \text{Er} @ \text{NaYF}_4: 20\% \text{Yb}$ structure is concerned, the authors did not observe increased contribution from the violet and red emission

to the overall emission either (Fig. 3b in the paper). The $\text{NaYF}_4@ \text{NaYbF}_4:2\% \text{Er}@ \text{NaYF}_4$ nanoparticles [Nature Photonics 12, 5488-553 (2018)] emit comparable green light and weaker violet and red light as compared with our nanostructures. Considering only one third of Er^{3+} ions exist in our trilayered structure ($\text{Er}/\text{Yb} \approx 1/133$ for $\text{C}_Y:10\% \text{Er}@ \text{S}_{Yb}@ \text{S}_Y$, $\text{Er}/\text{Yb} = 1/49$ for $\text{C}_Y@ \text{S}_{Yb}:2\% \text{Er}@ \text{S}_Y$) while the amounts of Yb^{3+} ions are nearly the same for both structures, we believe UC energy transfer efficiency of our structure is higher than the structure reported in Nature Photonics. In fact, the different variation tendencies of the lifetimes for the upconverted Er^{3+} emission bands with increasing Er^{3+} concentration in the two structures also validate higher efficiency of our trilayered structure (line 10-13 on page 10, Supplementary Fig. 16).

Figure 1. A comparison of $\alpha\text{-NaYF}_4:10\% \text{Er}@ \text{NaYbF}_4@ \text{NaYF}_4$ nanoparticles and the recently reported new nanostructures. (a-h) TEM images of $\alpha\text{-NaYF}_4@ \text{NaYb}_{0.98}\text{Er}_{0.02}\text{F}_4@ \text{NaYF}_4$ ($\text{C}_Y@ \text{S}_{Yb}: 2\% \text{Er}@ \text{S}_Y$) (a-c for the core, core-shell and core-shell-shell nanostructures), $\alpha\text{-NaYbF}_4@ \text{NaY}_{0.9}\text{Er}_{0.1}\text{F}_4@ \text{NaYbF}_4$ ($\text{C}_{Yb}@ \text{S}_Y: 10\% \text{Er}@ \text{S}_{Yb}$) (d-f for the core, core-shell and core-shell-shell nanostructures) and $\alpha\text{-NaYb}_{0.8}\text{Er}_{0.2}\text{F}_4@ \text{NaYF}_4$ ($\text{C}_{Yb}: 20\% \text{Er}@ \text{S}_Y$) (g, h for the core and core-shell nanostructures). (i) Upconversion emission spectra for these UCNPs upon 980-nm excitation ($24.0 \text{ W}/\text{cm}^2$).

In the meantime, we have amended the introduction (line 7-8 on page 3 and line 3-4 on page 4) and supplemented the corresponding references (17-18).

6) Some of the authors' names are not properly written or spelled. Please proofread the text for these minor errors.

Response: We have carefully checked the manuscript to avoid the minor errors.

Reviewer #3 (Remarks to the Author):

The authors have adequately addressed the comments and therefore i recommend acceptance of the manuscript.

Response: We thank the reviewer for positive evaluation.

Reviewers' Comments:

Reviewer #1:

Remarks to the Author:

In my opinion the authors made sufficient efforts to clarify the points raised by me and other reviewers. The article presents interesting new results and shines new light on the optimisation of UCNPs. There are some minor points to clarify before final acceptance:

1. Fig.2e - what is the difference between violet and brown samples (C_Y and C_{Yb}, respectively) - is that the Y or Yb present in the core? I found that information in Fig.S32, but most probably clear description is necessary in Fig.2e caption.
2. Fig.2d - the Y/Er content is high in α -NaYF₄: Er@NaYbF₄@NaYF₄, so why the composition map of Y is barely observable? Actually, the EDX maps are of not sufficient quality and does not confirm the core-shell structure in my opinion. The Fig.2b shows that indirectly, but does not confirm the real composition.
3. Fig.2f vs Fig.3d - Why there is no correspondence here? For optimum sample from Fig.2f (e.g. C_Y 10%Er@S_{Yb}@S_Y with relative integral intensity being 100-fold more efficient than C_Y 20%Yb2%Er@S_Y, one should expect to see similar ratios on Fig.3d for the 390-420 nm range - the same intermediate and outer shell thickness)
4. Fig.4d - the inset should show the numbers and give a chance to compare the intensities quantitatively
5. Fig.5c - has the SOSG fluorescence been calibrated to inform about the singlet oxygen content in mM - what I mean is - it is possible to say the C_Y-10%Er@S_{Yb}@S_Y sample produces larger amount of singlet oxygen than the other samples, but it is not clear is it a lot or not in terms of PDT efficiency as compared to conventional UV stimulated PDT. I know the direct PDT is more efficient in absolute terms, but when it comes to deep tissue PDT, the NIR activated PDT can overcome the direct-PDT, especially, when the QY of up-converting labels is reaching 15% (Fig.3b). This would be really great for NIR-PDT.
6. Fig.S15 - the graphs are vs Er concentration (2-70mol%) - so why 100% sample decay and NIR emission is shown as well?

Reviewer #2:

Remarks to the Author:

Evaluation:

- Data is technically sound and comprehensive: The data is now satisfactorily complete, sound, and comprehensive with the addition of key data points to shore up previous holes and weak points.
- Paper provides strong evidence: The manuscript contains strong evidence and rationale.
- Results are novel: Results are sufficiently novel.
- Manuscript is important to scientists in the specific field: Manuscript is sufficiently important to advancing the field.
- Recommendation: The novelty of the results and findings are sufficient and relevant enough to advancing the field of upconversion nanoparticles. Scientifically satisfactory for Nature Communications with minor improvements to the figure quality, as described below:
The authors have satisfactorily addressed the scientific questions and concerns of the reviews, and

they have appropriately accumulated data to further support their hypotheses. While the paper in its current state is scientifically acceptable, there remains room to improve:

- 1) Just a note, not a criticism: The comparative work done on the recently reported structures is highly valuable for this kind of work. I would like to note that in [Advanced Materials 31, 1806991 (2019)], the authors were correct to note that there is quenching after 2% doping of Er, but perhaps they should have done the comparisons using the 2% doping instead of 10% where there maybe quenching. Furthermore, the advanced materials paper did not use inert shell passivating layers which may also slightly compromise this comparison. That said, the comparisons overall are appreciated.
- 2) It would be appreciated if Figure 1 can be expanded on. As it stands it is a very weak figure/schematic. Perhaps an overall schematic including a short summary of the PDT application or an elaboration/expansion of the author's unique core-shell structure on an atomic/lattice scale.
- 3) The nanoparticle representations could be improved significantly. Perhaps you can use some 3d modeling software to quickly generation core-shell spheres. This can be done easily with software applications such as Blender, which are free and open-source, as well as other options including 3D Max or Rhino.
- 4) The figures overall can be polished further. While the overall layout is fine, some of the boxes can be aligned better etc. I would also request that the authors use the highest quality and resolution images for the figures. Some of the illustrations can be significantly improved.

Response to reviewers' comments

We thank the reviewers for their helpful comments to improve the paper as well as the quality of the figures. We modified the manuscript and redrew Figure 1-5 according to the comments. The comments are addressed accordingly below.

Reviewer #1 (Remarks to the Author):

In my opinion the authors made sufficient efforts to clarify the points raised by me and other reviewers. The article presents interesting new results and shines new light on the optimisation of UCNPs. There are some minor points to clarify before final acceptance:

Response: We are grateful for the reviewer's helpful comments about improving the manuscript.

1. Fig.2e - what is the difference between violet and brown samples (C_Y and C_Yb, respectively) - is that the Y or Yb present in the core? I found that information in Fig.S32, but most probably clear description is necessary in Fig.2e caption.

Response: The violet sample is designated as α -NaYF₄: 10%Er@NaYbF₄@NaYF₄ UCNPs, while the brown one represents α -NaYbF₄: 10%Er@NaYbF₄@NaYF₄ UCNPs. We have supplemented the information in the figure caption for clarity.

2. Fig.2d - the Y/Er/ content is high in α -NaYF₄: Er@NaYbF₄@NaYF₄, so why the composition map of Y is barely observable? Actually, the EDX maps are of not sufficient quality and does not confirm the core-shell structure in my opinion. The Fig.2b shows that indirectly, but does not confirm the real composition.

Response: We also agree with the reviewer that the EDX maps are difficult to differentiate core-shell structure accurately especially when the signal of Er is relatively weak. We tried to collect EDX maps for several times and the results are shown below. The results are very similar and the accurate border of the core is hard to discern. Because our nanoparticles are relatively smaller and unstable under electron irradiation, the collecting time is not long enough for us to acquire strong signals of each element on our present TEM facilities. We have

changed the dark blue color into bright yellow color to highlight Y elements in the revised Fig.2d. Moreover, we have supplemented the border lines of core and shells in the integrated map according to individual maps of Er, Yb and Y. To our knowledge, in addition to EDX mapping [Chu et al., Nat. Photon. 12, 548-553 (2018), Resch-Genger et al., Small 13, 1701635 (2017)], high-angle annular dark-field (HAADF) scanning TEM images are often used to observe core-shell UCNP structures [Wang et al. Nat. Commun. 10, 1811 (2019), Cohen et al. Nat. Commun. 9, 3082 (2018)]. In Supplementary Fig. 14a, one can clearly observe each layer in the composite structure except that the first shell is merged with the core due to similar compositions (NaYF₄ versus NaYF₄: 10%Er).

Figure 1-response EDX elemental mapping of core-shell-shell NaYF₄:10%Er@NaYbF₄@NaYF₄ UCNP.

3. Fig.2f vs Fig.3d - Why there is no correspondence here? For optimum sample from Fig.2f (e.g. C_Y 10%Er@S_{Yb}@S_Y with relative integral intensity being 100-fold more efficient than C_Y 20%Yb2%Er@S_Y, one should expect to see similar ratios on Fig.3d for the 390-420 nm range - the same intermediate and outer shell thickness)

Response: We have redrawn Fig. 3d, in which UCL intensity in the range of 390-420 nm for the samples with larger outer shell thickness (≥ 2.3 nm) is 1500-fold higher than C_Y:10%Er@S_{Yb} UCNP.

4. Fig.4d - the inset should show the numbers and give a chance to compare the intensities quantitatively

Response: For quantitative comparison, relative integral intensity of downshifting luminescence (versus integral intensity for UCNP with $d_{SYb} = 1.3$ nm) was shown in the inset of Fig.4d.

5. Fig.5c - has the SOSG fluorescence been calibrated to inform about the singlet oxygen content in mM - what I mean is - it is possible to say the CY-10%Er@S_{Yb}@S_Y sample produces larger amount of singlet oxygen than the other samples, but it is not clear is it a lot or not in terms of PDT efficiency as compared to conventional UV stimulated PDT. I know the direct PDT is more efficient in absolute terms, but when it comes to deep tissue PDT, the NIR activated PDT can

overcome the direct-PDT, especially, when the QY of up-converting labels is reaching 15% (Fig.3b). This would be really great for NIR-PDT.

Response: To our knowledge, the content of the singlet oxygen is difficult to acquire because there is no standard $^1\text{O}_2$ or SOSG endoperoxide samples (reaction product of SOSG with $^1\text{O}_2$). Only a $^1\text{O}_2$ quantum yield of a photosensitizer can be detected by using Rose Bengal (RB) with a high $^1\text{O}_2$ quantum yield of 0.76 as a model photosensitizer [Lin, et al. J. Fluoresc. 23, 41–47 (2013)]. Since the wavelength of excitation light (infrared light) in this work is quite different from that for RB, $^1\text{O}_2$ quantum yield of our UCNPs cannot be detected, either. We supplemented related information on page 15 (line 13-17) and the corresponding reference (ref. 37).

6. Fig.S15 - the graphs are vs Er concentration (2-70mol%) - so why 100% sample decay and NIR emission is shown as well ?

Response: We have corrected the figure caption by replacing (2-70mol%) with (2-100mol%).

Reviewer #2 (Remarks to the Author):

Evaluation:

- *Data is technically sound and comprehensive: The data is now satisfactorily complete, sound, and comprehensive with the addition of key data points to shore up previous holes and weak points.*
- *Paper provides strong evidence: The manuscript contains strong evidence and rationale.*
- *Results are novel: Results are sufficiently novel.*
- *Manuscript is important to scientists in the specific field: Manuscript is sufficiently important to advancing the field.*
- *Recommendation: The novelty of the results and findings are sufficient and relevant enough to advancing the field of upconversion nanoparticles. Scientifically satisfactory for Nature Communications with minor improvements to the figure quality, as described below:*

The authors have satisfactorily addressed the scientific questions and concerns of the reviews, and they have appropriately accumulated data to further support their hypotheses. While the paper in its current state is scientifically acceptable, there remains room to improve:

Response: We thank the reviewer for positive evaluation.

1) Just a note, not a criticism: The comparative work done on the recently reported structures is highly valuable for this kind of work. I would like to note that in

[Advanced Materials 31, 1806991 (2019)], the authors were correct to note that there is quenching after 2% doping of Er, but perhaps they should have done the comparisons using the 2% doping instead of 10% where there maybe quenching. Furthermore, the advanced materials paper did not use inert shell passivating layers which may also slightly compromise this comparison. That said, the comparisons overall are appreciated.

Response: We also noticed that the concentration quenching and passivating layers may compromise this comparison. To avoid misleading or confusing the readers, we emphasized that α -NaYF₄: 20%Yb@NaY_{0.98}Er_{0.02}F₄@NaYF₄: 20%Yb UCNPs possibly emit much stronger UCL than C_{Yb}@S_Y: 10%Er@S_{Yb} UCNPs in Supplementary Information. We also pointed out that higher doping concentration was adopted just to verify higher efficiency of our core-shell-shell structure in suppressing concentration quenching.

2) It would be appreciated if Figure 1 can be expanded on. As it stands it is a very weak figure/schematic. Perhaps an overall schematic including a short summary of the PDT application or an elaboration/expansion of the author's unique core-shell structure on an atomic/lattice scale.

Response: We have expanded Figure 1 with the corresponding lattice structures of core and shells supplemented, which highlights the structure more directly.

3) The nanoparticle representations could be improved significantly. Perhaps you can use some 3d modeling software to quickly generation core-shell spheres. This can be done easily with software applications such as Blender, which are free and open-source, as well as other options including 3D Max or Rhino.

Response: We redrew the core-shell spheres in Figures 3 and 5 using 3D Max for better representing the structure of the nanoparticles.

4) The figures overall can be polished further. While the overall layout is fine, some of the boxes can be aligned better etc. I would also request that the authors use the highest quality and resolution images for the figures. Some of the illustrations can be significantly improved.

Response: We redrew the figures to improve the quality. We also uploaded the original figures along with the revised manuscript.

Reviewers' Comments:

Reviewer #1:

Remarks to the Author:

I have no further questions and think the article can now be accepted, because it presents interesting new insights and ideas, and the quality of presentation is now acceptable.

The only comment is, that "förster resonance energy transfer (FRET)" should be started from capital letters - this forms the FRET acronym, and "Förster" is the family name of a scientist, and deserves such form of respect.

Reviewer #2:

Remarks to the Author:

- Data is technically sound and comprehensive: The data is now satisfactorily complete, sound, and comprehensive with the addition of key data points to shore up previous holes and weak points. Furthermore, the manuscript is now clearer, and the figures have been significantly improved.
- Paper provides strong evidence: The manuscript contains strong evidence and rationale.
- Results are novel: Results are sufficiently novel.
- The manuscript is important to scientists in the specific field: Manuscript is sufficiently important to advancing the field.
- Recommendation: The novelty of the results and findings are sufficient and relevant enough to advancing the field of upconversion nanoparticles for publication.

Response to reviewers' comments

We thank the reviewers for their precious comments. We modified the manuscript according to the comments.

Response to referees' comments

Reviewer #1

I have no further questions and think the article can now be accepted, because it presents interesting new insights and ideas, and the quality of presentation is now acceptable.

The only comment is, that "förster resonance energy transfer (FRET)" should be started from capital letters - this forms the FRET acronym, and "Förster" is the family name of a scientist, and deserves such form of respect.

Response: We would like to thank the reviewer for positive evaluation. We capitalized F in "Förster".

Reviewer #2

• Data is technically sound and comprehensive: The data is now satisfactorily complete, sound, and comprehensive with the addition of key data points to shore up previous holes and weak points. Furthermore, the manuscript is now clearer, and the figures have been significantly improved.

• Paper provides strong evidence: The manuscript contains strong evidence and rationale.

• Results are novel: Results are sufficiently novel.

• The manuscript is important to scientists in the specific field: Manuscript is sufficiently important to advancing the field.

• Recommendation: The novelty of the results and findings are sufficient and relevant enough to advancing the field of upconversion nanoparticles for publication.

Response: We thank the reviewer for positive evaluation.